# Mesenchymal Tumors of the Liver: An Update Review

**DOI:** 10.3390/biomedicines13020479

**Published:** 2025-02-15

**Authors:** Joon Hyuk Choi, Swan N. Thung

**Affiliations:** 1Department of Pathology, Yeungnam University College of Medicine, 170 Hyeonchung-ro, Namgu, Daegu 42415, Republic of Korea; 2Department of Pathology, Molecular and Cell-Based Medicine, Icahn School of Medicine at Mount Sinai, 1468 Madison Avenue, New York, NY 10029, USA; swan.thung@mountsinai.org

**Keywords:** mesenchymal tumors, liver, hemangioma, angiosarcoma, mesenchymal hamartoma

## Abstract

Hepatic mesenchymal tumors (HMTs) are non-epithelial benign and malignant tumors with or without specific mesenchymal cell differentiation. They are relatively uncommon. Except for mesenchymal hamartoma, calcified nested stromal–epithelial tumor, and embryonal sarcoma, most mesenchymal lesions are not specific to the liver. Pathologists face challenges in diagnosing HMTs due to their diverse morphologies and phenotypic variations. Accurate diagnosis is critical for directing appropriate patient care and predicting outcomes. This review focuses on mesenchymal tumors with a relative predilection for the liver, including vascular and non-vascular mesenchymal neoplasms. It provides a thorough and up-to-date overview, concentrating on clinical and pathological features, differential diagnosis, and diagnostic approaches.

## 1. Introduction

Hepatic mesenchymal tumors (HMTs) are a group of non-epithelial benign and malignant tumors with specific mesenchymal cell differentiation or a lack of specific differentiation [1]. HMTs are a diverse group of soft tissue tumors that are relatively uncommon in the liver. Most mesenchymal lesions of the liver are not unique to this organ. However, some mesenchymal tumors are specific to the liver, such as mesenchymal hamartoma (MH), calcifying nested stromal–epithelial tumor (CNSET), and embryonal sarcoma (ES). Table 1 lists the relatively common and rare HMTs.

In the fifth edition of the World Health Organization (WHO) classification of digestive system tumors [2,3], there is a chapter on mesenchymal tumors. New subtypes and entities in the liver have been identified, including inflammatory angiomyolipoma [4], anastomosing hemangioma [5], diffuse hemangiomatosis, and hepatic small vessel neoplasm (HSCN) [6]. In addition, important molecular data have been included for several tumors. For example, epithelioid hemangioendothelioma (EHE) is linked to the *WWTR1*::*CAMTA1* fusion gene and the *YAP1*::*TFE3* fusion gene [7,8], whereas CSNET is linked to *CTNNB1* mutations [9].

Accurate pathological diagnosis and understanding of HMTs’ molecular pathogenesis are critical for effective patient management and prognosis. However, diagnosis and treatment of HMTs remains challenging due to their diverse histological and molecular genetic profiles. This review provides a comprehensive and current overview of the pathology of HMTs, with a focus on advances in histological subtypes, molecular genetics, differential diagnosis, and diagnostic approaches.

## 2. Adipocytic Tumors

### 2.1. Lipoma

#### 2.1.1. Clinical Features

Lipoma is a benign neoplasm made of mature adipocytes [10]. Hepatic lipomas are uncommon and generally asymptomatic. They usually occur in adults aged 40 to 60 years and are typically discovered incidentally during imaging studies such as ultrasonography, computed tomography (CT), and magnetic resonance imaging (MRI). Like soft tissue lipomas, hepatic lipomas are more prevalent in obese people and strongly associated with hepatic steatosis [11].

#### 2.1.2. Pathological Features

Hepatic lipomas can range in size from a few millimeters to several centimeters. They are clearly defined lesions surrounded by a thin, fibrous capsule. They have a uniform yellow appearance on the cut surface and a soft, oily texture. Histologically, lipomas consist of mature adipocytes with minimal size variation. Areas of fat necrosis or hemorrhage may occasionally be observed. The adipocytes have peripheral, flattened nuclei and no atypia. Lipomatous tumors are classified based on the addition of tissue elements to the adipose component. These include fibrolipoma, myxolipoma, angiolipoma, chondrolipoma, osteolipoma, and myelolipoma [1].

#### 2.1.3. Differential Diagnosis

Differential diagnosis includes pseudolipoma and focal fatty nodule [12,13]. Pseudolipomas should be distinguished from hepatic lipomas. Pseudolipomas are typically present as small, encapsulated fat masses located within concavities on the liver surface and often exhibit necrosis and calcification [12]. In contrast, hepatic lipomas are found in the hepatic parenchyma. Focal fatty nodules, also known as focal fatty change or focal fatty liver, can sometimes be mistaken for lipomatous lesions. They consist of nodular aggregates of hepatocytes with macrovesicular steatosis that retain the acinar architecture [13].

### 2.2. Liposarcoma

#### 2.2.1. Clinical Features

Although liposarcomas, malignant neoplasms exhibiting adipocytic differentiation, are the most common sarcomas in adults, hepatic liposarcomas are extremely rare [14]. Several cases of hepatic liposarcoma have been reported in the literature [15,16,17]. Primary hepatic liposarcoma affects both men and women equally, and the clinical presentations are typically nonspecific. Symptoms may comprise abdominal distension, epigastric mass, and weight loss. Hepatic liposarcomas have the same histological subtypes as soft tissue liposarcomas [18,19,20]. The prognosis is determined by the histological subtype and grade of the tumor.

#### 2.2.2. Pathological Features

Liposarcoma usually presents a large, well-defined, lobulated mass with a yellow-to-white cut surface. Liposarcomas are classified into four histological subtypes: (1) well differentiated, (2) dedifferentiated, (3) myxoid, and (4) pleomorphic. Well-differentiated liposarcoma is characterized by variation in adipocyte size, atypical hyperchromatic nuclei of adipocytes and stromal cells, and up-regulation of *MDM2* and/or *CDK4*. Dedifferentiated liposarcoma is characterized by an abrupt or gradual transition from well-differentiated liposarcoma to a low- or high-grade spindle cell or pleomorphic, non-lipogenic tumor (rarely lipogenic). The majority of dedifferentiated liposarcomas show amplification of *MDM2* and *CDK4*. Myxoid liposarcoma consists of uniform, round-to-ovoid cells, scattered with lipoblasts, and delicate arborizing capillaries within a myxoid stroma. Pathognomonic genetic alterations include the *FUS*::*DDIT3* fusion gene and, less commonly, the *EWSR1*::*DDIT3* fusion gene. Pleomorphic liposarcoma is a pleomorphic spindle cell sarcoma containing varying numbers of pleomorphic lipoblasts. Immunohistochemically, well-differentiated and dedifferentiated liposarcomas are positive for MDM2 and CDK4. Myxoid liposarcomas exhibit DDIT3 expression.

#### 2.2.3. Differential Diagnosis

Differential diagnosis includes lipoma and metastatic liposarcoma. Lipomas are distinguished from well-differentiated liposarcomas by the lack of significant adipocyte size variation and cytological atypia in adipocytes or stromal cells. Furthermore, lipomas are negative for MDM2 expression. Metastatic liposarcomas should be excluded during the initial evaluation [21,22]. In the liver, metastatic liposarcomas are often present as multiple lesions. A prior history of a primary liposarcoma in an extraperitoneal site, particularly the retroperitoneum or extremities, strongly suggests a metastatic spread.

## 3. Fibroblastic and Myofibroblastic Tumors

### 3.1. Inflammatory Myofibroblastic Tumor

#### 3.1.1. Clinical Features

Inflammatory myofibroblastic tumor (IMT) is a fibroblastic/myofibroblastic neoplasm with intermediate biological potential, characterized by a significant inflammatory cell infiltrate consisting predominantly of lymphocytes and plasma cells [23]. The liver is an uncommon site for IMT [24]. Although it predominantly affects children and young adults, cases have been reported from a wide age range [25]. Common symptoms at presentation include abdominal pain, abdominal discomfort, fever, and weight loss [26]. The cause remains unknown. Tyrosine kinase receptor gene rearrangements, particularly those involving the *ALK* locus at 2p23 and various fusion partners, occur in two-thirds of IMTs, [27]. Epithelioid inflammatory myofibroblastic sarcoma (EIMS) is an aggressive IMT subtype that commonly manifests as widespread intra-abdominal masses [28,29,30]. EIMS harbors the *RANBP2*::*ALK* fusion gene or *RRBP1*::*ALK* fusion gene. Approximately 5% of IMTs harbor *ROS1* gene fusions. Rare cases of IMTs have shown gene fusions involving *NTRK3*, *PDGFRB*, and *RET* [31,32]. Conventional gastrointestinal IMTs have low local recurrence rates [26], with distant metastases occurring in <5% of cases [33]. However, ALK-negative tumors are more likely to metastasize.

#### 3.1.2. Pathological Features

Most tumors range in size from 3 cm to 12 cm and have firm white or yellow cut surfaces. Histologically, the tumors consist of uniform, plump spindle cells with vesicular chromatin, small nucleoli, and pale cytoplasm, with a loose fascicular pattern [25]. The stroma is myxoid or collagenous, with an inflammatory infiltrate composed primarily of lymphocytes and plasma cells, with occasional eosinophils and neutrophils. A subset of tumor cells may have ganglion cell-like characteristics. Mitotic activity is low. Necrosis may occasionally be present. EIMS consists of epithelioid or round cells with prominent nucleoli and amphophilic or eosinophilic cytoplasm, accompanied by a marked neutrophilic infiltrate [28]. Immunohistochemically, the tumor cells are positive for smooth muscle actin (SMA) in almost all cases and desmin in more than half of them. Cytokeratins expression is observed in 20–30% of cases. Approximately 60% of IMTs express ALK in a diffuse cytoplasmic pattern [34], while approximately 5% express ROS1, corresponding to the underlying *ALK* and *ROS1* gene fusions, respectively [31]. EIMSs exhibit distinctive nuclear membrane or perinuclear ALK staining patterns [28].

#### 3.1.3. Differential Diagnosis

Differential diagnosis includes inflammatory pseudotumor, follicular dendritic cell sarcoma (FDCS), and leiomyosarcoma. The term “inflammatory pseudotumor” was previously used interchangeably with IMT due to their similar histological and clinical characteristics. However, it is a benign, non neoplastic reactive lesion with varying amounts of fibrous tissue, fibroblasts, myofibroblasts, and chronic inflammatory cells, primarily lymphocytes and plasma cells [1]. Inflammatory pseudotumors exhibit less organized fibroblast/myofibroblast proliferation and no *ALK* or *ROS1* rearrangements [35]. FDCS is a malignant neoplasm with the morphological and phenotypic characteristics of follicular dendritic cells [36]. The majority of hepatic FDCS cases are classified as Epstein–Barr virus (EBV)-positive inflammatory follicular dendritic cell sarcoma (EFDCS) [37]. EFDCS is characterized by the neoplastic proliferation of FDCs, a marked lymphoplasmacytic infiltrate, and its association with EBV [38,39]. EDFCSs are positive for EBV-encoded small RNA (EBER) and express FDC markers such as CD21, CD23, and CD35. Leiomyosarcomas can present with prominent inflammation and are characterized by cigar-shaped nuclei with blunted ends and brightly eosinophilic cytoplasm. Unlike IMTs, leiomyosarcomas are negative for ALK.

### 3.2. Solitary Fibrous Tumor

#### 3.2.1. Clinical Features

Solitary fibrous tumor (SFT) is a fibroblastic neoplasm characterized by distinctive thin-walled, branching, staghorn-shaped vessels and the *NAB2*::*STAT6* fusion gene [40]. SFTs primarily affect adults aged 20 to 70 years, with rare cases involving children and adolescents. These tumors can develop in virtually any anatomical location [41,42,43,44]. In the liver, SFTs are thought to originate from the Glisson capsule. Although most SFTs have solitary lesions, they can occasionally appear multifocal [45,46,47,48,49]. Symptoms are usually nonspecific; however, hypoglycemia can occur if the tumor produces insulin-like growth factor 2 [50]. Molecularly, SFTs are defined by the *NAB2*::*STAT6* fusion gene, which results from an inversion at the 12q13 locus [51,52,53]. Prognostication of SFT is notoriously challenging. Although most SFTs are benign, approximately 10% exhibit aggressive characteristics, resulting in local recurrence and distant metastasis [40].

#### 3.2.2. Pathological Features

Hepatic SFTs can appear as intraparenchymal or exophytic pedunculated masses that range in size from 2 cm to 20 cm. Most SFTs are solitary, but they can be multifocal. Grossly, the tumors have a light tan to white-gray cut surface with a whorled appearance. Histologically, SFTs are composed of bland, ovoid-to-spindle-shaped cells with pale eosinophilic cytoplasm. These cells exhibit a random, patternless pattern, and the presence of thin-walled, branching, or staghorn-shaped blood vessels is a distinguishing feature. The level of cellularity varies, even within the same tumor. Stromal and perivascular hyalinization is commonly observed. Indicators of an increased risk of aggressive behavior include older patient age, larger tumor size, high cellularity, cytological atypia, >4 mitoses/2 mm^2^, necrosis, and sarcomatous transformation [54,55,56]. Multivariate risk models have increased prognostic accuracy beyond the limitations of traditional benign and malignant classification [56]. Immunohistochemically, SFTs are positive for CD34 and STAT6 [57]. STAT6 is a highly sensitive and specific marker for diagnosing SFT [58].

#### 3.2.3. Differential Diagnosis

Differential diagnosis includes perivascular epithelioid cell tumor (PEComa), dedifferentiated liposarcoma, leiomyosarcoma, and metastatic SFT. Fat-forming (lipomatous) SFTs can resemble PEComas, including angiomyolipoma. PEComas are distinguished by the expression of smooth muscle markers (e.g., SMA, h-caldesmon, desmin) and melanocytic markers (e.g., HMB-45, melan-A, MITF). Rare cases of dedifferentiated liposarcomas may have SFT-like morphology and are characterized by MDM2 positivity. Leiomyosarcomas consist of intersecting fascicles of spindle cells with blunt-ended nuclei and eosinophilic fibrillary cytoplasm. Unlike SFTs, they are negative for STAT6. Metastatic SFTs are significantly more common than primary SFTs, so careful clinical correlation is required for a correct differential diagnosis.

## 4. Vascular Tumors

### 4.1. Hepatic Congenital Hemangioma

#### 4.1.1. Clinical Features

Hepatic congenital hemangioma (HCH) is a benign vascular neoplasm that is fully developed at birth and characterized as GLUT1-immunonegative [59]. There are three clinical subtypes: rapidly involuting CH (RICH), partially involuting (PICH), and non-involuting CH (NICH). In most cases, involution occurs during infancy [59]. HCHs are typically asymptomatic and are often detected incidentally during infancy imaging or prenatal ultrasonography [60,61,62]. In some cases, newborns may exhibit hepatomegaly, thrombocytopenia, and mild hypofibrinogenemia. In rare cases, these conditions may progress to cardiac failure as a result of arteriovenous shunting [63,64]. HCH is more common in girls and can be associated with cutaneous infantile hemangioma or cavernous hemangioma [60,62]. The majority of RICH cases are caused by missense mutations that change glutamine at amino acid position 209 (Q209), leading to hyperactivation of the *GNAQ* and *GNA11* genes [65]. A subset of the cases are linked to *PIK3CA* mutations [66].

#### 4.1.2. Pathological Features

HCHs are typically solitary lesions [61], with multiple foci being uncommon. Grossly, HCHs are well-circumscribed, soft-to-firm masses that appear tan to white (Figure 1a). They range in size from a few millimeters to more than 10 cm. Central hemorrhage and necrosis may be seen. Histologically, HCHs are composed of aggregates or lobules of capillary-like channels lined by plump-to-hobnailed endothelial cells and surrounded by pericytes (Figure 1b). As HCH involutes, the vascular structures are gradually replaced by fibrous stroma. The intervascular stroma is fibrotic, with entrapped hepatocytes, bile ducts scattered throughout, extramedullary hematopoiesis foci, and occasionally dilated sinusoids. Central necrosis, dystrophic calcification, chronic inflammatory cells, macrophages (sometimes containing hemosiderin), and dysplastic veins with thrombi are common findings. Immunohistochemically, the lesion’s endothelial cells within the lesion are positive for vascular markers and WT1 [67] but negative for GLUT1 [68].

#### 4.1.3. Differential Diagnosis

Differential diagnosis includes hepatic infantile hemangioma (HIH) [69]. Unlike HCHs, HIHs typically appear within a few weeks after birth. Immunohistochemically, the endothelial cells in HIHs are positive for GLUT1.

### 4.2. Hepatic Infantile Hemangioma

#### 4.2.1. Clinical Features

HIH is a benign, proliferative, vascular neoplasm characterized by endothelial GLUT1 immunopositivity [70]. It is the most common HMT in children [71]. HCH and HIH account for approximately 12% of all pediatric liver tumors [59]. Most HIHs are asymptomatic and discovered incidentally during imaging tests. Infants with HIH are more likely to be female, have a low birth weight, arrive prematurely, and have multiple gestations [72,73,74,75]. Many patients are asymptomatic. When present, symptoms are nonspecific and may include nausea, lethargy, or gastrointestinal bleeding. They may be associated with hepatoblastoma, MH [76,77,78,79], and Beckwith–Wiedemann syndrome. HIH has a mortality rate of up to 16%. The majority of these children present with diffuse HIH, abdominal compartment syndrome, or cardiac failure [80]. If the proliferative phase continues after the age of 2, the risk of progression to angiosarcoma increases [81,82]. Aggressive forms of HIH have been linked to the overexpression of hedgehog signaling pathway components such as sonic hedgehog, GLI2, and their target gene *FOXA2* [83].

#### 4.2.2. Pathological Features

HIH is typically multifocal or diffuse and rarely represents a solitary lesion. The diffuse pattern is characterized by innumerable, frequently coalescent nodules that can extensively replace the hepatic parenchyma [61]. Individual nodules range in size from a few millimeters to more than 10 cm, appearing as well-defined, tan-white, oval-to-round, and spongy masses. Hemorrhage and necrosis may be present. Histologically, HIHs are composed of anastomosing, sinusoid-like channels with flattened or minimally plump endothelial cells, particularly in the central portions of the nodules. These cells have small, bland, hyperchromatic nuclei. Occasionally, endothelial cells form pseudopapillary architectures with increased cellularity. The stroma is fibrous, and entrapped bile ducts and hepatocytes may be seen. Individual masses are separated by the normal hepatic parenchyma. Immunohistochemically, the endothelial cells of the lesions are positive for GLUT1 and vascular markers such as CD31 and CD34.

#### 4.2.3. Differential Diagnosis

Differential diagnosis includes HCH [84], MH, and angiosarcoma. HCHs are fully developed at birth after forming during fetal development in utero. Unlike HIHs, HCHs are negative for GLUT1. MHs may occasionally have a prominent vascular component [78,85], which can mimic HIH. However, MHs are distinguished by the presence of both epithelial and stromal components and thick-walled vessels with round lumina. Angiosarcomas are distinguished by marked cytological atypia and frequent mitotic activity. In cases of HIH, moderate to severe endothelial atypia, solid spindle or epithelioid foci, necrosis, and variability in endothelial GLUT1, expression may indicate progression to angiosarcoma.

### 4.3. Cavernous Hemangioma

#### 4.3.1. Clinical Features

Hemangiomas are benign vascular neoplasms [86]. According to autopsy studies, cavernous hemangiomas are among the most common benign HMTs, accounting for approximately 20% of the general population [87,88]. They can occur at any age, but are most often observed in women between their third and fifth decades. Cavernous hemangiomas are typically asymptomatic and are frequently found incidentally during imaging studies. However, lesions larger than 4 cm may become symptomatic, resulting in pain or bleeding [86]. Although they are generally solitary, multiple lesions can occasionally occur. During pregnancy, cavernous hemangiomas can grow or rupture. They may also grow or recur in patients receiving estrogen therapy, suggesting that both endogenous and exogenous estrogen exposure may play a role in the development and progression of these tumors [89]. Hepatic capillary hemangiomas have been rarely reported in adults [90,91].

#### 4.3.2. Pathological Features

Cavernous hemangiomas are well-defined, purple-red lesions with a spongy consistency (Figure 2a). However, in cases of severe sclerosis, they may appear white and firm. These lesions vary significantly in size [92]. Histologically, cavernous hemangiomas consist of dilated vascular channels lined by a single layer of flattened endothelial cells and fibrous septa of varying thickness (Figure 2b). Organizing thrombi, infarction, fibrosis, and calcification may be observed. Lesions larger than 10 cm are called giant cavernous hemangiomas [93]. Fibrosis is common, particularly in the center of the tumor, and it is thought to be caused by a regressive process. These cases are frequently referred to as sclerosing cavernous hemangiomas [94]. In some cases, extensive fibrosis may completely obscure cavernous architecture, leading to the designation of sclerosed hemangioma [94]. Diffuse hepatic hemangiomatosis (DHH) is an extremely rare disease characterized by extensive replacement of the liver parenchyma by diffuse multiple hemangiomas. In some cases, other organs may also be affected [86]. Immunohistochemically, the endothelial lining of cavernous hemangiomas consistently expresses vascular markers such as CD31, CD34, and FLI1.

#### 4.3.3. Differential Diagnosis

Differential diagnosis includes peliosis hepatis, hereditary hemorrhagic telangiectasia (HHT), angiosarcoma, and fibrotic liver lesions. In most cases, cavernous hemangiomas are easily diagnosed. Bacillary angiomatosis is a vascular proliferation occurring in immunocompromised individuals and is caused by opportunistic infection by *Bartonella* species. When it involves the liver, it manifests as peliosis hepatis [95]. Unlike cavernous hemangiomas, peliosis hepatis presents cystic blood-filled spaces surrounded by hepatocytes and lacks an endothelial lining and fibrous septa. DHHs must be distinguished from HHTs [86]. HHTs are distinguished by dilated vascular channels in the portal and periportal areas, which are frequently accompanied by aberrant portal vessels. DHHs should not be misinterpreted as angiosarcoma based solely on their infiltrating appearance. Angiosarcomas are distinguished by atypical endothelial cells, anastomosing vascular channels, and multilayered endothelial cell proliferation. Sclerosed hemangiomas should be differentiated from other fibrotic liver lesions, including MHs, bile duct adenomas, and intrahepatic cholangiocarcinomas. Immunohistochemical staining for endothelial markers can help identify the remaining vascular network.

### 4.4. Anastomosing Hemangioma

#### 4.4.1. Clinical Features

Anastomosing hemangioma is a rare benign vascular neoplasm characterized by complex, anastomosing vascular channels lined by bland endothelial cells [87]. Although it is most commonly described in the genitourinary tract, it has also been reported less frequently in the liver and gastrointestinal tract [96,97]. Anastomosing hemangiomas are typically asymptomatic and are discovered incidentally in adults, with a higher prevalence in women. This tumor exhibits mutations in *GNAQ*, *GNA11*, and *GNA14* [98,99].

#### 4.4.2. Pathological Features

Anastomosing hemangioma is well demarcated. Histologically, it consists of anastomosing capillary-like vessels with a loosely lobular architecture. The endothelial cells are arranged in a single layer and occasionally exhibit a hobnail appearance. Mild endothelial atypia can be observed, but the tumor lacks infiltrative growth into the liver parenchyma. Areas resembling cavernous hemangioma are also seen. Intracytoplasmic hyaline globules may also be present in the neoplastic endothelial cells. Fibrin thrombi are commonly found within vascular spaces. Approximately half of the tumors exhibit extramedullary hematopoiesis. Immunohistochemically, the endothelial cells are positive for vascular markers such as CD31, CD34, and ERG.

#### 4.4.3. Differential Diagnosis

Differential diagnosis includes angiosarcoma. Anastomosing hemangiomas may resemble angiosarcomas due to their anastomosing growth pattern. However, anastomosing hemangiomas are typically well-circumscribed, lack significant cytological atypia, and do not show diffuse infiltration into the surrounding liver parenchyma [5].

### 4.5. Hepatic Small Vessel Neoplasm

#### 4.5.1. Clinical Features

HSVN is a rare vascular neoplasm considered benign or low-grade [86]. HSVN was initially identified as a lesion with uncertain biological potential. However, long-term clinical follow-up has shown that HSVN is generally benign but can exhibit locally aggressive behavior [100,101]. HSVN primarily affects adults, with an average patient age of 54 years (range, 24–83 years) [87]. It occurs more commonly in men. These tumors are typically asymptomatic and are often incidentally discovered during imaging tests as single liver masses. Molecularly, HSVN is similar to anastomosing hemangioma, with mutations in *GNAQ*, *GNA11*, and *GNA14* [100]. Given these similarities, anastomosing hemangioma and HSVM may represent variations in the same entity. Recently, molecular risk stratification for the malignant potential in HSVN has been proposed [102].

#### 4.5.2. Pathological Features

HSVNs are typically small lesions, averaging 2.1 cm in diameter, and are often poorly defined. Histologically, HSVN is characterized by small, thin-walled vessels with flattened-to-plump hobnail endothelial cells that contain erythrocytes within the lumen [86]. These vessels can infiltrate around the portal tracts and between the hepatic plates. Extramedullary hematopoiesis is occasionally observed. The adjacent hepatic parenchyma may show the hepatocyte plate’s expansion, with areas that resemble focal nodular hyperplasia. Immunohistochemically, the lining epithelium expresses vascular markers such as CD31, CD34, and ERG.

#### 4.5.3. Differential Diagnosis

Differential diagnosis includes cavernous hemangioma, anastomosing hemangioma, and angiosarcoma. Cavernous hemangiomas are typically well-circumscribed and distinguished by large vascular spaces lined by flat, uniform endothelial cells. Although HSVNs resemble anastomosing hemangiomas, they are differentiated by their infiltrative interface with the surrounding hepatic parenchyma, whereas anastomosing hemangiomas are clearly demarcated from the liver parenchyma. HSVNs can mimic hepatic angiosarcoma due to their infiltrative nature; however, they lack cytological atypia, multilayering, mitotic activity, and necrosis [6]. Additionally, HSVNs have a Ki-67 proliferation index of less than 10%, whereas angiosarcomas have an index greater than 10% [6].

### 4.6. Lymphangioma

#### 4.6.1. Clinical Features

Lymphangioma is a benign tumor consisting of lymphatic spaces lined by a single layer of lymphatic endothelial cells and filled with serous or chylous material [103]. The term “lymphangiomatosis” refers to multicentric or extensively infiltrating lymphangiomas. Hepatic lymphangiomas are rare, with most documented cases occurring as part of diffuse lymphangiomatosis [104]. These lesions are usually present at birth or in early childhood. Lymphangiomas are thought to result from developmental malformations. Growth factors, such as vascular endothelial growth factor C, vascular endothelial growth factor receptor-3, and Prox-1, play a role in their pathogenesis [105]. Somatic mutations in the *PIK3CA* gene have also been found to contribute to their development [106].

#### 4.6.2. Pathological Features

Hepatic lesions can be solitary or multiple lesions, and frequently appearing multiloculated. These lesions are cystic and contain either clear or chylous, milky fluid [107]. Histologically, they are characterized by a single layer of attenuated lymphatic endothelial cells lining dilated cystic spaces with no cytological atypia. Some of these spaces are partially enclosed by smooth muscle and/or fibromuscular walls. Clear fluid or foamy histiocytes may be present in the lymphatic spaces, and scattered lymphoid aggregates are common in the surrounding stroma. Immunohistochemically, the endothelial cells are positive for CD31 and D2-40 (podoplanin), with variable expression of CD34. The smooth muscle cells surrounding the cystic spaces are positive for SMA.

#### 4.6.3. Differential Diagnosis

Differential diagnosis includes lymphangiectasia, hemangioma, and angiosarcoma. Lymphangiectasia is distinguished by the dilatation of native lymphatic vessels, whereas lymphangiomas exhibit lymphatic vessel proliferation and are more likely to present as nodules or masses. Hemangiomas are composed of smaller vascular spaces with a higher concentration of red blood cells, less proteinaceous material, and fewer lymphocytes [108]. Angiosarcomas are distinguished by their atypical vascular proliferation and irregular, anastomosing vascular spaces. Endothelial cells in angiosarcomas exhibit nuclear atypia and multilayering.

### 4.7. Kaposi Sarcoma

#### 4.7.1. Clinical Features

Kaposi sarcoma (KS) is a vascular neoplasm linked to human herpes virus 8 (HHV8). It is characterized by disorganized endothelial cell proliferation, leading to erythrocyte-containing clefts, organized neovascularization, and accompanying inflammatory infiltrates [109]. Four clinical and epidemiological forms are known: (1) classic indolent KS, (2) endemic African KS, (3) acquired immunodeficiency syndrome (AIDS)-associated KS, and (4) iatrogenic KS. KS is detected in approximately 15% of acquired immunodeficiency syndrome (AIDS) patients. Disseminated hepatic KS is most commonly observed in association with AIDS. Some hepatic KS cases may be associated with iatrogenic factors, such as immunosuppression following transplantation. Advanced KS lesions are often oligoclonal and contain viral episomes of varying sizes, suggesting that KS may be a reactive or non-neoplastic proliferation rather than a true neoplasm with metastatic behavior [110]. While most patients with hepatic KS have no symptoms, rare cases can rapidly progress to clinically significant disease, resulting in multiorgan failure and liver failure, which are often fatal [111]. Molecularly, KS is characterized by recurrent 11q13 gains with *FGF4* and *INT2* up-regulation, clonal Y chromosome loss in early disease, and chromosomal copy number changes (16, 17, 21, X, and Y) in late-stage lesions [112,113].

#### 4.7.2. Pathological Features

KS typically affects the portal and periportal regions of the liver, with lesions appearing as reddish-brown foci. Histologically, KS is characterized by an infiltrative proliferation of small, irregular vascular channels and fascicles of relatively uniform spindle cells with mild atypia. Slit-like vascular spaces and cytoplasmic eosinophilic hyaline globules are present. The extravasation of red blood cells, lymphoplasmacytic infiltrates, and hemosiderin deposition are common features. Immunohistochemically, the endothelial and spindle cells are positive for CD31, CD34, ERG, and D2-40 (podoplanin). The nuclear expression of HHV8 is a defining feature of the tumor. HHV8 can be identified in endothelial and spindle cells using polymerase chain reaction and in situ hybridization [114].

#### 4.7.3. Differential Diagnosis

Differential diagnosis includes angiosarcoma and leiomyosarcoma. Angiosarcomas are characterized by anastomosing, infiltrative vascular channels lined with highly atypical endothelial cells, and often have a multilayered structure. They are negative for HHV8. Leiomyosarcomas are distinguished by intersecting fascicles of atypical smooth muscle cells and are negative for HHV8 and vascular markers.

### 4.8. Epithelioid Hemangioendothelioma

#### 4.8.1. Clinical Features

EHE is a rare malignant vascular neoplasm composed of epithelioid cells in a fibrous or myxohyaline stroma and linked to either a *WWTR1*::*CAMTA1* or *YAP1*::*TFE3* fusion gene [115]. EHE is sporadic and has a slight female predominance, most frequently affecting middle-aged adults, although it can occur at any age [116]. Hepatic EHE may manifest jaundice, abdominal pain, and ascites. Multifocal liver involvement has been found in >75% of patients [116,117,118]. Approximately 70% of patients have elevated alkaline phosphatase levels [118]. More than 90% of EHE cases have a t(1;3)(p36;q25) translocation, resulting in the *WWTR1*::*CAMTA1* fusion gene [119,120,121]. A subset of EHEs (approximately 5%) harbors a *YAP1*::*TFE3* fusion gene [8,122]. The clinical course varies, ranging from progressive disease in some patients to indolent, stable disease in others [9]. The distant metastatic rate is approximately 20–30% [116,117].

#### 4.8.2. Pathological Features

EHEs range in size from small subcentimeter nodules to large masses greater than 10 cm. The cut surface is yellow to white, firm, and often has multifocal nodules. Histologically, EHE shows an infiltrative or variable nodular growth pattern with a prominent myxohyaline or fibrous stroma. The tumors consist of epithelioid cells with fine chromatin, small nucleoli, and a moderate amount of eosinophilic cytoplasm. Intracytoplasmic vacuoles are occasionally observed in tumor cells and may contain erythrocytes (Figure 3a). The tumor cells are arranged either singly or in cords. An invasion of sinusoids, portal veins, and hepatic veins is frequently observed and is sometimes accompanied by necrosis. EHEs with *YAP1*::*TFE3* fusion have a brightly eosinophilic cytoplasm, a higher likelihood of solid growth patterns, and frequent formation of vascular spaces. Immunohistochemically, the tumor cells are positive for vascular markers, such as CD31, CD34, factor VIII-related antigen, FLI1, and ERG (Figure 3b) [123]. CAMTA1 expression is observed in cases with the *WWTR1*::*CAMTA1* fusion gene, corresponding to an underlying gene rearrangement [124,125]. Transcription factor E3 (TFE3) positivity is seen in tumors harboring the *YAP1*::*TFE3* fusion gene. Epithelial antigens such as cytokeratins 7, 8, and 18 are expressed in approximately 40% of cases [126].

#### 4.8.3. Differential Diagnosis

Differential diagnosis of EHE includes epithelioid angiosarcoma, intrahepatic cholangiocarcinoma, and metastatic carcinoma. Epithelioid angiosarcomas have more significant nuclear atypia and frequent mitoses, and they are negative for CAMTA1 [125]. EHEs frequently express cytokeratins [127], which can lead to a misdiagnosis as intrahepatic cholangiocarcinoma or metastatic carcinoma. However, intrahepatic cholangiocarcinoma and metastatic carcinomas are negative for vascular markers. A thorough patient history, particularly primary tumors, is essential for a correct differential diagnosis.

### 4.9. Angiosarcoma

#### 4.9.1. Clinical Features

Angiosarcoma is a malignant vascular neoplasm that exhibits endothelial differentiation and can form a broad spectrum of vascular structures [128]. Angiosarcoma is the most common malignant HMT in adults, accounting for approximately 2% of all primary hepatic cancers [129]. Clinical symptoms are usually nonspecific, and patients may present with fatigue, weight loss, abdominal pain, a palpable mass, or ascites [130]. While adult hepatic angiosarcoma occurs spontaneously [131], most pediatric cases develop within pre-existing HIH [132,133,134,135,136,137]. Risk factors for hepatic angiosarcoma include exogenous exposure to chemical carcinogens, such as vinyl chloride [138,139,140], androgens [141], cyclophosphamide [142], diethylstilbestrol, and arsenic [130], as well as exposure to thorium oxide (Thorotrast) [143], radium, and radiation. Genetically, angiosarcomas exhibit genetic heterogeneity. Most angiosarcomas harbor complex genomic profiles [144]. Hepatic angiosarcomas are frequently associated with ATRX deficiency and can show alternative lengthening of telomeres phenotype [145]. Approximately 10% of sporadic AS cases, particularly in younger patients, harbor various types of *CIC* mutations or fusions [146]. Most hepatic ASs are highly aggressive, and survival times of more than a year are uncommon [130,132]. Factors associated with poor prognosis include older age, large tumor size, and a high Ki-67 proliferation index [147,148].

#### 4.9.2. Pathological Features

Hepatic angiosarcomas are typically poorly defined and have varying appearances (Figure 4a,b). They range from solid grayish-white tissue to markedly hemorrhagic areas. Histologically, the tumor demonstrates a broad morphological spectrum, ranging from well-formed, anastomosing vessels to solid sheets of epithelioid or spindle cells (Figure 4c). Multilayering, hobnailing, and intraluminal papillary tufts may also be observed. Nuclear atypia, atypical mitoses, and necrosis are usually present. The major growth patterns are classified as mass-forming and sinusoidal (non-mass forming) types [149,150]. The sinusoidal growth pattern is characterized by patchy sinusoidal dilatation lined with atypical cells displaying enlarged nuclei (Figure 5a,b). Epithelioid angiosarcoma exhibits a solid pattern and is characterized by atypical epithelioid or polygonal cells with vesicular nuclei, prominent nucleoli, and abundant cytoplasm. Immunohistochemically, angiosarcomas exhibit membranous positivity for CD31 and nuclear positivity for ERG and variable positivity for CD34 (~50%), factor VIII-related antigen, FLI1, and D2-40 (podoplanin) [151,152]. Cytokeratin and epithelial membrane antigen (EMA) expressions may be seen, particularly in epithelioid angiosarcomas [153]. Immunohistochemistry (IHC) for p53 and MYC are helpful in identifying sinusoidal-type angiosarcomas [150,154].

#### 4.9.3. Differential Diagnosis

Differential diagnosis includes anastomosing hemangioma, HSVN, EHE, metastatic angiosarcoma, and carcinoma. Anastomosing hemangiomas and HSVNs lack endothelial multilayering or tufting. The presence of a p53 mutant staining pattern or a Ki-67 proliferation index greater than 10% are strong markers of angiosarcoma [6]. EHEs have a distinct myxohyaline stroma and positivity for CAMTA1, and lack the well-developed vasoformation seen in angiosarcoma [124,125]. Primary hepatic angiosarcomas must be distinguished from metastatic angiosarcoma of the liver [128]. A thorough clinical history and radiological findings are required for a correct differential diagnosis. Cytokeratin and EMA expressions may be observed in epithelioid angiosarcomas, leading to misdiagnosis as metastatic carcinoma. However, metastatic carcinomas lack vasoformative properties and are negative for vascular markers.

## 5. Pericytic (Perivascular) Tumors

### 5.1. Glomus Tumor

#### 5.1.1. Clinical Features

Glomus tumor is a neoplasm composed of cells similar to the perivascular modified smooth muscle cells found in the normal glomus body [155]. It can rarely occur in the liver, with most tumors appearing as solitary lesions, though up to 10% of patients may have multiple lesions. Hepatic glomus tumors typically develop in adults between the fourth and seventh decades of life, have a male predominance, and are larger than their cutaneous counterparts [156,157,158,159]. Clinical symptoms often include epigastric fullness and pain. Multiple familial glomus tumors are caused by inactivating mutations in the glomulin gene (*GLMN*) and are inherited in an autosomal dominant pattern. Approximately half of benign and malignant GTs from various sites exhibit recurrent rearrangements in NOTCH family genes, often involving the fusion of *NOTCH1/2/3* and *MIR143* [160]. *BRAF* p.V600E mutations have also been identified in some sporadic glomus tumors, which may be linked to aggressive behavior. BRAF represents a potential therapeutic target for patients with progressive disease [161]. Most gastrointestinal glomus tumors are benign. However, malignant glomus tumors have been reported in a few cases [155].

#### 5.1.2. Pathological Features

Grossly, glomus tumors are well-circumscribed and often appear as multinodular masses. Cystic changes and calcifications may be observed. Histologically, the tumors consist of uniform, round cells with central, round nuclei and moderate amounts of clear to eosinophilic cytoplasm. A sharply defined basal lamina encloses individual cells. The stroma may be hyalinized or myxoid, and tumor cells may occasionally exhibit oncocytic or epithelioid cytomorphology [162,163]. Focal nuclear atypia and vascular invasion are relatively frequent findings in glomus tumors. However, vascular involvement is not associated with adverse prognosis. In the liver, the criteria for malignancy remain undefined due to limited data. In the peripheral soft tissue, criteria for malignancy include a deep location and size > 2 cm, moderate-to-high nuclear grade, atypical mitotic figures, and ≥5 mitoses/10 mm^2^ [164]. Immunohistochemically, glomus tumors show diffuse and strong expression of SMA in nearly all cases and h-caldesmon in more than 60%. Most tumors also exhibit pericellular net-like positivity for collagen type IV and laminin [165]. Cytoplasmic Sirtuin 1 (SIRT1) expression has also been documented [166].

#### 5.1.3. Differential Diagnosis

Differential diagnosis includes neuroendocrine tumor (NET) and metastatic glomus tumor. Glomus tumors can look very similar to NETs. Focal synaptophysin expression may occur in glomus tumors, posing a diagnostic challenge [165]. However, NETs have a speckled or “salt-and-pepper” chromatin pattern, are positive for cytokeratin, and are negative for SMA. Primary hepatic glomus tumors should be distinguished from metastatic tumors that originate in other sites. A complete patient history, including the presence of prior glomus tumors or glomus tumors in different body locations, is required for an accurate diagnosis.

## 6. Smooth Muscle Tumors

### 6.1. Leiomyoma

#### 6.1.1. Clinical Features

Leiomyoma is a benign neoplasm with smooth muscle differentiation [167]. Primary hepatic leiomyomas are extremely rare [10,168,169,170,171]. They have been observed in pediatric and adult populations ranging in age from 5 to 87 years, with a slight preference for females [169]. The criteria for diagnosing a primary hepatic smooth muscle tumor are as follows: (1) the tumor must comprise leiomyocytes, and (2) there must be no other leiomyomatous tumors anywhere else [172].

#### 6.1.2. Pathological Features

Leiomyomas are well-circumscribed, unencapsulated tumors with a gray-white, whorled cut surface. Their size differs widely, ranging from subcentimeter to 20 cm. Histologically, primary hepatic leiomyomas exhibit the same morphology as leiomyomas arising in the other parts of the body. Most leiomyomas consist of short fascicles of spindle-shaped, well-differentiated smooth muscle cells. These spindle cells are uniform, with blunt-ended nuclei, abundant eosinophilic cytoplasm, and no cytological atypia. Fibrosis, myxoid change, calcification, and degenerative changes may be observed. Immunohistochemically, the tumor cells show positivity for SMA, h-caldesmon, and desmin.

#### 6.1.3. Differential Diagnosis

Differential diagnosis includes PEComa and leiomyosarcoma. PEComas are distinguished by epithelioid and spindle cells with granular eosinophilic or clear cytoplasm. Immunohistochemically, PEComas exhibit positivity for melanocytic markers such as HMB-45 and melan-A. Leiomyosarcomas are differentiated by nuclear atypia, increased mitotic activity, and necrosis.

### 6.2. Epstein–Barr Virus-Associated Smooth Muscle Tumor

#### 6.2.1. Clinical Features

EBV-associated smooth muscle tumor (ESMT) is a smooth muscle neoplasm caused by EBV infection that typically affects patients with immunosuppression [173]. The majority of cases occur in one of three major settings: HIV-associated immunodeficiency, immunodeficiency following solid organ or hematopoietic stem cell transplantation, and congenital or primary immunodeficiency [174,175]. The pathogenesis of ESMTs is driven by EBV infection, combined with T lymphocyte immunosuppression. These tumors can arise in various anatomical locations, but they most commonly affect the liver. The multicentric nature is thought to be caused by multiple independent EBV infection events rather than metastatic spread [173]. MYC overexpression and AKT/mTOR pathway activation are critical in EBV-induced smooth muscle tumor proliferation [176,177]. The patient’s immune status has a significant impact on their prognosis. The majority of these tumors do not metastasize [173].

#### 6.2.2. Pathological Features

ESMTs vary in size from less than 1 cm to more than 20 cm [178,179,180]. The cut surface is white to gray with a rubbery-to-firm consistency, and the margins may be either well defined or infiltrative. Histologically, the tumor consists of intersecting fascicles of spindle-shaped cells with elongated nuclei and abundant eosinophilic cytoplasm. A second population of round, more primitive-appearing smooth muscle cells can be observed in approximately half of the cases. A subset of tumors exhibits a hemangiopericytoma-like pattern. Cytological atypia is typically mild to moderate, but it can be severe in HIV-positive patients, whose tumors often show increased mitotic activity and/or necrosis. Intratumoral T lymphocytic infiltrates are commonly present [178]. Immunohistochemically, the tumor cells are diffusely positive for smooth muscle markers such as SMA and h-caldesmon, whereas desmin typically shows focal expression. EBER in situ hybridization is consistently positive.

#### 6.2.3. Differential Diagnosis

Differential diagnosis includes leiomyoma and leiomyosarcoma. Leiomyomas are histologically similar to ESMT in most cases, but they lack primitive round cells and are negative for EBER in situ hybridization. Leiomyosarcomas have nuclear atypia and pleomorphism, increased mitotic activity, and necrosis, and are negative for EBER in situ hybridization.

### 6.3. Leiomyosarcoma

#### 6.3.1. Clinical Features

Leiomyosarcoma is a malignant neoplasm that exhibits smooth muscle differentiation [181]. Primary hepatic leiomyosarcoma is rare and typically arises in the intrahepatic vascular structures, muscular walls of the bile ducts, or ligamentum teres [182,183,184,185]. It affects both men and women equally, but cases involving the inferior vena cava are more common in women. The majority of cases occur in adults, but some cases have been reported in immunosuppressed children. The clinical manifestations are primarily nonspecific and may include weight loss, anorexia, vomiting, jaundice, abdominal pain, and, in rare cases, intra-abdominal bleeding caused by tumor rupture. Tumors arising in the hepatic veins can cause Budd–Chiari syndrome. Genetically complex aneuploid karyotypes are found in the majority of cases. Inactivation of *TP53* and *RB1* is often seen [186,187]. In the digestive system, leiomyosarcomas are aggressive neoplasms, with a local recurrence rate of 40–80% and a metastasis rate of 55–70% [181]. The disease-specific mortality rate ranges from 25% to 50%, depending on the tumor site.

#### 6.3.2. Pathological Features

Leiomyosarcomas are typically large and solitary tumors. The cut surface is fleshy, pinkish-white, and rubbery-to-firm. Areas of necrosis and hemorrhage are frequently observed. Histologically, the tumor resembles leiomyosarcomas in other anatomical sites. It consists of elongated spindle cells arranged in intersecting fascicles. Tumor cells have elongated, cigar-shaped, blunt-ended nuclei and brightly eosinophilic cytoplasm. They are arranged in fascicular, palisaded, or hemangiopericytoma-like patterns. Nuclear atypia and hyperchromasia, necrosis, and increased mitotic activity are seen. Histological subtypes include myxoid, epithelioid, pleomorphic, and dedifferentiated leiomyosarcomas. Immunohistochemically, the tumor cells usually show diffuse cytoplasmic positivity for SMA and variable expression for h-caldesmon and desmin. Additionally, some tumors may show focally positivity for cytokeratins and EMA.

#### 6.3.3. Differential Diagnosis

Differential diagnosis includes leiomyoma, PEComa, metastatic leiomyosarcoma, and metastatic gastrointestinal stromal tumor (GIST). Leiomyosarcomas are distinguished from benign leiomyomas by nuclear atypia, increased mitotic activity, and necrosis. PEComas are distinguished by epithelioid, spindled, or ovoid cells with clear to granular eosinophilic cytoplasm. These tumors are positive for smooth muscle markers (e.g., SMA, desmin) and melanocytic markers (e.g., HMB-45, melan-A). Metastatic leiomyosarcomas are more common than primary hepatic leiomyosarcomas. Therefore, a thorough examination of the patient’s medical history and clinical and imaging findings is required to differentiate them from primary tumors. Metastatic GISTs show positivity for CD117 and DOG-1, whereas leiomyosarcomas are negative for CD1117 and DOG1.

## 7. Skeletal Muscle Tumors

### 7.1. Rhabdomyosarcoma

#### 7.1.1. Clinical Features

Rhabdomyosarcoma (RMS) is a malignant neoplasm with skeletal muscle differentiation to varying degrees [188]. RMSs are extremely rare in the biliary tract. However, the botryoid subtype of embryonal RMS is the most common type, particularly in children. This type accounts for 1% of all pediatric RMS cases [189]. Most patients are under the age of 5, with a median age of 3 years [190,191]. However, cases have also been reported in older children and adults [192,193]. Lesions are rare at birth. Males are more frequently affected, with a male-to-female ratio of approximately 2:1 [193]. The typical clinical presentation includes obstructive jaundice, abdominal distention, fever, or loss of appetite [190]. Genetically, embryonal RMSs are characterized by gains of chromosomes 2, 8, 11, 12, and 13, along with mutations in the RAS pathway (identified in 50% of cases) and TP53 mutations (occurring in 10%) [194]. Additionally, loss of heterozygosity at 11p15.5 is a common occurrence. Other subtypes of RMS, such as alveolar and pleomorphic RMS, have rarely been reported in the liver [195,196].

#### 7.1.2. Pathological Features

Grossly, the botryoid subtype appears as soft, tan masses extending along the biliary tree, often protruding into the lumen in a typical botryoid growth pattern [197]. The walls of the affected ducts are typically thickened. Histologically, RMS subtypes exhibit distinct features. The botryoid subtype of embryonal RMS has a cambium layer formed by tumor cell condensation beneath the epithelial surface, distributed in a loose fibromyxoid stroma. The tumor cells are round, stellate, or spindle-shaped, with hyperchromatic nuclei and some mitotic figures. Cross-striations may also be observed. The botryoid masses are covered by unremarkable biliary epithelium with occasional focal erosion. The alveolar RMS consists of primitive, round tumor cells arranged in an alveolar growth pattern. The solid subtype of alveolar RMS is distinguished by solid sheets of tumor cells that exhibit the cytomorphological characteristics of alveolar RMS. Pleomorphic RMS is a high-grade sarcoma characterized by pleomorphic polygonal, spindle-shaped cells, or round cells. Immunohistochemically, all RMSs show relatively strong positivity for desmin, myogenin, and MYOD1. Myogenin expression is typically diffuse in alveolar RMS, but it can be focal or patchy in embryonal RMS.

#### 7.1.3. Differential Diagnosis

Differential diagnosis includes leiomyosarcoma, ES, and extrarenal rhabdoid tumor (ERT). Leiomyosarcomas lack the cambium layer and rhabdomyoblasts, and are negative for myogenin and MYOD1. ESs typically have prominent nuclear pleomorphism and cytoplasmic hyaline globules. Myogenin and MYOD1 are particularly useful markers for distinguishing RMS from ES, as ESs can express SMA and desmin but do not express myogen or MYOD1 [198]. ERTs are distinguished by their larger cells with eccentric nuclei and hyaline globular cytoplasmic inclusions. They are positive for cytokeratins and EMA but negative for desmin and myogenin and show a loss of nuclear SMARCB1 (integrase interactor 1 [INI1]) expression.

## 8. Tumors of Uncertain Differentiation

### 8.1. Mesenchymal Hamartoma

#### 8.1.1. Clinical Features

MH is a benign tumor characterized by a multicystic mass of loose connective tissue with a bile duct component, often exhibiting features of ductal plate malformation [199]. MH is the third most common hepatic tumor in childhood, following hepatoblastoma and HIH. MH accounts for approximately 12% of all liver tumors diagnosed within the first 2 years of life. Approximately 85% of MHs occur before the age of 3 years, while <5% of MHs are diagnosed after the age of 5 years [200]. MH exhibits a slight male predominance. Symptoms may include an enlarging abdomen and a nontender mass. The cause of MH remains unknown; however, these tumors can occur in association with Beckwith–Wiedemann syndrome [201]. Cytogenetic and molecular genetic evidence indicates that MH is a neoplastic entity rather than a developmental anomaly [202]. Genetically, chromosomal rearrangements involving 19q13.4 are common and are known to activate the chromosome 19 microRNA cluster (C19MC) [203]. Liver function is usually normal in MH patients, although serum alpha-fetoprotein (AFP) levels may be slightly elevated [204]. MH is typically benign, with an excellent prognosis if the mass is completely resected [205]. However, several cases of progression to undifferentiated ES have been reported [206,207].

#### 8.1.2. Pathological Features

Most MHs show expanding, well-circumscribed, white-to-yellow masses without a capsule [1] (Figure 6a). The cut surface typically reveals multiple cystic spaces that do not communicate with the bile ducts in approximately 85% of cases. In very young patients, MHs have a more solid appearance and fewer cysts, indicating that cyst formation occurs progressively as the tumor grows. In a study of 17 MH cases, 41% of MHs were solid, and 59% were cystic [208]. The cysts range in size from a few millimeters to 15 cm and contain yellow fluid or gelatinous material. Histologically, MHs consist of loose connective tissue and bile ducts in various proportions arranged in lobulated patterns [199] (Figure 6b). The stromal component is typically loose and myxoid, high in glycosaminoglycans. It contains bland spindle- or stellate-shaped fibroblasts and myofibroblasts, dilated blood vessels, and fluid-filled cyst spaces with no endothelial lining. The bile ducts are often branched, tortuous, or dilated and commonly exhibit a pattern of ductal plate malformation. Entrapped islands of hepatocytes may also be present. Non-epithelial-lined cysts form within the mesenchyme due to fluid accumulation, and foci of extramedullary hematopoiesis, are observed in >85% of cases.

#### 8.1.3. Differential Diagnosis

Differential diagnosis includes HIH, lymphangioma, and mucinous cystic neoplasm. Focally regressed HIHs may share some overlapping characteristics with MHs. However, the vascular elements in HIHs are derived from smaller vessels, resulting in thinner and less muscular vessel walls. Additionally, an anastomosing capillary network may still be present in regressed HIHs. Multiple cysts without an epithelial lining may be misinterpreted as lymphangioma. However, lymphangiomas are far less fibrotic than MH and do not contain torturous biliary elements. Lymphangiomas are positive for vascular markers. Mucinous cystic neoplasms are distinguished from MH by the presence of both mucinous epithelium and ovarian-type stroma.

### 8.2. Perivascular Epithelioid Cell Tumor (PEComa)/Angiomyolipoma

#### 8.2.1. Clinical Features

PEComa is a mesenchymal neoplasm composed primarily of epithelioid cells that express smooth muscle and melanocytic markers [209]. PEComas are generally classified into two main groups: (1) angiomyolipomas (AMLs) and (2) non-angiomyolipoma PEComas. AML is a subtype of PEComa that typically arises in the kidney and liver. It is composed of varying proportions of blood vessels, smooth muscle, and adipose tissue, leading to triphasic morphology. Non-angiomyolipoma PEComas can arise in various locations, including soft tissue, the uterus, the lungs, and the gastrointestinal tract. Unlike AMLs, they lack triphasic histologic features and are often composed predominantly of spindle or epithelioid cells with perivascular accentuation. These tumors share the same perivascular epithelioid cell lineage as AML. Hepatic PEComa most commonly affects middle-aged adults, with a significant female predominance [210,211,212]. PEComas are usually found incidentally. Large lesions may present with abdominal pain. Although most hepatic PEComas occur sporadically, 5–10% of cases are linked to tuberous sclerosis [211,213]. PEComas are frequently associated with *TSC2* or *TSC1* mutations (biallelic inactivation), which activate the mTOR signaling pathway [214,215,216,217]. Additionally, a subset of PEComas exhibit *TFE3* gene rearrangements [214,218]. The majority of hepatic PEComas are benign [210,211,212]. However, malignant hepatic PEComas have rarely been reported [219]. Alterations of *TP53*, *RB1*, *ATRX*, *APC*, and *NF1* are observed in patients with most metastatic renal epithelioid AMLs, but they are rare in patients with nonmetastatic disease [220].

#### 8.2.2. Pathological Features

Hepatic PEComas are usually solitary, but occasionally multiple tumors can occur, particularly in patients with tuberous sclerosis. Their size varies widely [210,211]. The cut surface often appears yellow-to-tan and fleshy, with areas of necrosis and hemorrhage (Figure 7a). Histologically, PEComas consist primarily of epithelioid tumor cells with granular eosinophilic to clear cytoplasm (Figure 7b). The tumor cells are arranged in a nested, sheet-like, alveolar or trabecular pattern, accompanied by a delicate capillary vascular network. Most hepatic AMLs consist of a mixture of adipose tissue, epithelioid cells resembling smooth muscle cells, and thick-walled blood vessels in varying proportions. Two distinct subtypes of PEComa are recognized based on histological features. The inflammatory subtype of PEComa is characterized by a prominent chronic inflammatory cell infiltrate (Figure 8a,b) [4]. The sclerosing subtype of PEComa is characterized by tumor cells arranged in cords or trabeculae within a dense collagenous stroma. Malignant PEComas of soft tissue are characterized by a variable combination of nuclear pleomorphism, mitotic activity, and necrosis [221,222,223]. However, the criteria for malignancy in hepatic PEComa have not yet been validated. Immunohistochemically, PEComas are positive for both smooth muscle markers (e.g., SMA, h-caldesmon, desmin) and melanocytic markers (e.g., HMB45, melan-A, tyrosinase, MITF) [224,225,226] (Figure 7c,d). Approximately 15% of the PEComas demonstrate nuclear staining for TFE3, particularly in the *TFE3*-rearranged subtypes [218,224].

#### 8.2.3. Differential Diagnosis

Differential diagnosis includes hepatocellular carcinoma (HCC), leiomyosarcoma, fat-containing SFT, and metastatic GIST. PEComas are sometimes mistaken for HCCs due to the presence of epithelioid polygonal cells. However, HCCs can be distinguished by their expression for hepatocellular markers, such as Hep Par-1 and arginase-1. PEComas composed primarily of spindle cells may mimic leiomyosarcomas. However, leiomyosarcomas lack adipose tissue and are negative for melanocytic markers. Fat-containing SFTs may also mimic AMLs, but they can be differentiated by their positivity for CD34 and STAT6. PEComas may occasionally exhibit focal positivity for CD117 [224,227] and should not be confused with metastatic GISTs. Metastatic GISTs lack adipose tissue and are negative for melanocytic markers.

### 8.3. Calcifying Nested Stromal–Epithelial Tumor

#### 8.3.1. Clinical Features

CNSET is a rare, low-grade hepatic neoplasm of uncertain differentiation, with a distinct nested architecture surrounded by a cellular myofibroblastic stroma and psammomatous calcifications [228]. CNSET is extremely rare, with approximately 40 reported cases to date. It primarily affects children, adolescents, and young adults, with a female predominance [229,230]. CNSET is often discovered incidentally; some patients may have a history of calcified hepatic nodules. Most CNSETs occur sporadically, but some cases have been linked to Beckwith–Wiedemann syndrome [231,232,233]. Deletions in exon 3 of the *CTNNB1* gene have been identified [9]. A recent study found *TERT* promoter mutations and *CTNNB1* alterations in CNSETs, indicating that these tumors may have a more aggressive clinical course than previously reported [234]. The majority of patients with CNSET are cured through surgical excision. However, some cases have demonstrated locally recurrent disease, necessitating liver transplantation [235].

#### 8.3.2. Pathological Features

CNSETs are well circumscribed and lobulated. Tumor sizes range from 2.8 cm to 30 cm, with the majority measuring more than 10 cm. The cut surface is yellow or white and granular [230]. Histologically, CNSET is composed of nests of spindled to epithelioid cells surrounded by bands of stroma. The tumor cells are bland and uniform, with vesicular chromatin, indistinct nucleoli, and eosinophilic cytoplasm. The stroma is either loose and edematous or densely collagenous and desmoplastic. Some nests exhibit central necrosis. Psammomatous calcifications are frequently present within the nests and occasionally accompanied by ossification. Small bile duct-like structures can be seen around the edges of the tumor nests. Immunohistochemically, the epithelial tumor cells are positive for broad-spectrum cytokeratins. WT1 expression is typically nuclear, but may also present as cytoplasmic or in a perinuclear dot-like pattern. Aberrant nuclear and cytoplasmic β-catenin staining is observed [229,230,233]. The tumor cells may express CD56, EMA, progesterone receptor, neuron-specific enolase, and CD117. The stromal spindle cells are positive for SMA.

#### 8.3.3. Differential Diagnosis

Differential diagnosis of CNSET includes hepatoblastoma, HCC, and neuroendocrine tumor (NET). Unlike CNSET, the fetal subtype of hepatoblastoma lacks a dense fibrous stroma and consistently expresses hepatocellular markers such as Hep Par-1 and arginase-1 [236]. Compared to CNEST, HCCs exhibit more cytological atypia and express hepatocellular markers. CNSETs can show morphologic overlaps with NETs [101]. However, CNSETs are consistently negative for neuroendocrine markers such as synaptophysin and chromogranin A, whereas NETs are consistently positive for these markers.

### 8.4. Embryonal Sarcoma of the Liver

#### 8.4.1. Clinical Features

ES of the liver, also called undifferentiated ES, is a malignant mesenchymal neoplasm with diverse morphology and no specific differentiation [237]. It is the most common malignant HMT in the pediatric population [238]. ES typically affects children between the ages of 5 and 15, with no sex predilection [239,240], although rare cases also occur in adults [241]. Patients often present with abdominal distension, palpable mass, pain, fever, and weight loss. Elevated serum alkaline phosphatase and leukocytosis may be present. ES usually arises in the right hepatic lobe [242,243]. The combination of C19MC hyperexpression resulting from chromosomal structural alterations and *TP53* mutation or loss is a frequently recurrent genomic feature of ES [244]. Chromosomal rearrangements involving 19q13.4 are shared with MH [237]. Given their shared cytogenetic abnormalities, MH and ES are often thought to be part of the same disease spectrum [245]. The transformation from MH to ES has also been documented [246].

#### 8.4.2. Pathological Features

ESs are well circumscribed but unencapsulated. These tumors range from 10 cm to 30 cm in size. The cut surface is heterogeneous, with solid, fleshy, myxoid, and variable-sized cystic areas, often accompanied by necrosis and hemorrhage [239]. Histologically, ES consists of undifferentiated spindle, stellate, and pleomorphic tumor cells (Figure 9a). The stroma is usually myxoid, but it can also be dense and fibrous. The tumor cells show significant cytological atypia, and multinucleated giant, bizarre tumor cells are often present. The tumor cells are arranged in loose, solid, fascicular, or storiform patterns [239,247]. Periodic acid–Schiff (PAS)-positive, diastase-resistant intracytoplasmic eosinophilic hyaline globules are characteristically present [247] (Figure 9b). Entrapped bile ducts, hepatocytes, and extramedullary hematopoiesis are often present. Immunohistochemically, ES lacks a distinct immunophenotype. The tumor cells express diffuse vimentin and α1-antitrypsin. Variable and limited expressions of cytokeratins, desmin, SMA, and glypican-3 may also be observed [247,248,249].

#### 8.4.3. Differential Diagnosis

Differential diagnosis includes MH, embryonal RMS, high-grade pleomorphic sarcomas, and sarcomatoid HCC. MHs are composed of both epithelial and stromal components. The epithelial component consists of clustered hepatocytes and branched bile ducts, while the stromal component includes fibroblasts, inflammatory cells, and small blood vessels. Embryonal RMSs of the biliary tract are distinguished by a botryoid growth pattern within the bile duct lumen. They have rhabdomyoblasts with cross striations, a subepithelial cambium layer, express skeletal muscle markers such as myogenin and MYOD1 [250]. Primary and metastatic high-grade pleomorphic sarcomas are extremely rare in children. Immunohistochemical staining is critical in distinguishing these tumors. Sarcomatoid HCCs are predominantly characterized by spindle cells with focal areas of HCC. These tumors are focally positive for hepatocellular markers such as Hep Par-1 and arginase-1.

### 8.5. Extrarenal Rhabdoid Tumor

#### 8.5.1. Clinical Features

ERT, also known as malignant rhabdoid tumor, is a rare and highly malignant soft tissue tumor composed of undifferentiated tumor cells with a rhabdoid cytomorphologic appearance [107]. The tumor is characterized by mutations or deletions of the *SMARCBl* (*INI1*) gene. The liver is the single most common visceral site of involvement [251,252]. The majority of patients present within the first 3 years of life, with a median age of diagnosis at 8 months. Rare cases have been reported in adults [253,254,255]. Men and women are affected equally. Common symptoms include abdominal mass or distention, fever, vomiting, and pain, with tumor rupture occurring frequently [252]. Laboratory tests show normal or slightly elevated serum AFP levels [252]. Metastatic disease is detected at the initial presentation in approximately 76% of patients, with the most common sites of metastasis being the lungs, lymph nodes, central nervous system, skin, and bone [256]. Regardless of where the tumor is located, the prognosis is very poor.

#### 8.5.2. Pathological Features

Most tumors are solid and measure more than 5 cm in their greatest diameter. The cut surface typically appears gray-to-tan and is often accompanied by necrosis and hemorrhage. Multiple nodules may also be present. Histologically, ERT consists of small to medium-sized round cells with vesicular nuclei and distinct nucleoli. The cells are arranged in solid sheets and show infiltrative growth without an obvious fibrous capsule. Cystic degeneration or a trabecular architecture may be seen. Some tumor cells have distinct paranuclear spherical inclusions composed of intracellular filamentous aggregates, which contribute to their rhabdoid cytomorphology [1]. Cytoplasmic eosinophilic inclusions are PAS-positive. Immunohistochemically, most tumors express both mesenchymal (e.g., vimentin) and epithelial (e.g., cytokeratins, EMA) markers. The tumor cells show a loss of SMARCB1 (INIl) expression or SAMRCA4 expression [257]. SALL4 and glypican-3 expressions are frequently detected [258,259].

#### 8.5.3. Differential Diagnosis

Differential diagnosis includes RMS, PEComa, and small cell undifferentiated hepatoblastoma. While rhabdoid cells are characteristic of ERTs, they are not entirely specific, as similar cells can be found in other tumor types. ERTs are considered a diagnosis of exclusion. RMSs are positive for desmin and myogenin while retaining SMARCB1 (INIl). PEComas typically express melanocytic markers such as HMB45 and melan-A and are negative for cytokeratins and EMA. The similarity of clinicopathologic features between ERTs and small cell undifferentiated hepatoblastomas can cause diagnostic challenges [260]. Small cell undifferentiated hepatoblastomas typically retain SMARCBl (INI1) expression. Tumors exhibiting SMARCB1 (INI1) nuclear staining are now classified as ERTs [261,262].

Table 2 summarizes the clinical, histological, immunohistochemical, and molecular characteristics of selected HMTs.

## 9. Miscellaneous Mesenchymal Tumors of the Liver

Other rare mesenchymal tumors of the liver include myelolipoma [263], chondroma [264], schwannoma [265], neurofibroma [266], malignant peripheral nerve sheath tumor [267], osteosarcoma [268], desmoplastic small round cell tumor [269], and undifferentiated pleomorphic sarcoma [270]. Each of these tumors has distinct clinicopathological characteristics and genetic findings.

## 10. Diagnostic Approach

### 10.1. Clinical and Radiological Evaluation

A systematic approach is required for an accurate diagnosis of liver lesions. A thorough examination of the patient’s age, gender, and clinical and radiologic data provides valuable diagnostic insights. HMTs commonly found in infants and children include MH, HCH, HIH, embryonal RMS, ES, and ERT. In contrast, HMTs that preferentially affect adults include cavernous hemangioma, EHE, angiosarcoma, and PEComa [107]. HMTs are much less common than epithelial tumors. Therefore, it is critical to rule out non-mesenchymal tumors, such as carcinoma, malignant melanoma, germ cell tumor, or malignant lymphoma, before diagnosing a mesenchymal tumor [182,271]. Metastatic tumors appear in the liver more commonly than primary hepatic sarcomas. Multiple liver masses generally indicate metastasis, although exceptions may occur. Various imaging techniques, such as ultrasonography, CT, and MRI, are essential for assessing HMTs. Correlation with radiological imaging is particularly important when small biopsies are obtained, because it allows for a comprehensive and accurate assessment.

The characteristic radiological findings are helpful for the differential diagnosis of selected HMTs. HIH typically appears as a hypoechoic or complex mass on ultrasonography [272]. Non-enhanced CT scans reveal a low-attenuation mass, with fine calcifications present in approximately 50% of cases. MRI is particularly useful for assessing multifocality. Arteriography may reveal enlarged, tortuous feeding arteries. Cavernous hemangiomas are isodense to large blood vessels and exhibit characteristic cloud-like peripheral enhancement on CT. EHE is characterized by low-attenuation peripheral tumor nodules on CT. MRI often demonstrates a signal halo around the nodules. Angiosarcomas present as hypodense masses on nonenhanced CT, which become isodense on delayed postcontrast scans. Angiography may reveal an abnormal vascular pattern. MH appears to be a complex multicystic mass on ultrasonography. CT imaging reveals variability in cyst size and septal thickness. AML appears as a heterogeneous, well-circumscribed, hyperechoic lesion on ultrasound. On CT, it presents as a hypodense mass with marked early contrast enhancement and delayed enhancement in the portal venous phase. MRI is highly sensitive for detecting the fatty component, which exhibits high signal intensity on T1-weighted images. ES is characterized by a hypodense mass with solid and cystic areas on CT. It is typically hypovascular, distinguishing it from other vascular tumors.

### 10.2. Histological Evaluation

A careful pathological evaluation is required for an accurate diagnosis of liver lesions. First and foremost, a thorough gross examination is essential. The tumor number, size, location, color, consistency, and circumscription of all lesions should be meticulously evaluated. The distance from all lesions to the surgical resection margin should be measured. The microscopic examination begins with a detailed evaluation of hematoxylin-eosin-stained sections at low magnification. This initial review should concentrate on key features such as tumor cell morphology (e.g., spindle, round, epithelioid, and pleomorphic cells) (Table 3), growth pattern (e.g., fascicular, nested, lobular, vasoformative), and stromal characteristics (e.g., collagenous, inflammatory, and myxoid stroma). Certain tumors have distinctive nuclear characteristics (e.g., cigar-shaped, blunt-ended nuclei in leiomyosarcoma), cytoplasmic traits (e.g., clear or eosinophilic granular cytoplasm in PEComa), and stromal characteristics (e.g., prominent inflammatory cell infiltration in IMT and inflammatory PEComa). If available, reviewing slides from the original tumor is critical for confirming or excluding metastasis.

### 10.3. Immunohistochemical and Molecular Evaluation

Many mesenchymal tumors lack distinguishing morphological features and have an uncertain line of differentiation. IHC plays a critical role in determining the line of differentiation and often serves as a surrogate marker for underlying molecular genetic alterations [266]. Useful immunohistochemical markers for HMTs include vascular markers (e.g., CD31, CD34, factor VIII-related antigen, FLI1, ERG), smooth muscle markers (e.g., SMA, h-caldesmon, desmin), skeletal muscle markers (e.g., myogenin, MYOD1), and melanocytic markers (e.g., HMB-45, melan-A, MITF), and epithelial markers (e.g., cytokeratins, EMA) (Table 4). MDM2, ALK, and STAT6 can be useful in differential diagnosis. To ensure diagnostic accuracy, immunohistochemical results should always be interpreted in the context of a comprehensive panel, considering all relevant clinical and pathological information. Specific molecular analyses may be needed to diagnose definitively when histological features and immunohistochemical findings are inconclusive. The most accurate diagnosis of HMTs requires a multimodal approach that includes clinical history, radiological findings, tumor morphology, IHC, and molecular testing.

## 11. Treatment of Hepatic Mesenchymal Tumors

Treatment strategies of HMTs vary depending on tumor type, size, growth pattern, and malignancy potential. Small, asymptomatic hemangiomas (≤5 cm) require no treatment. However, surgical resection is recommended for larger hemangiomas (>10 cm) and when patients present with symptoms such as pain, bleeding, or compression of adjacent structures. For hepatic sarcomas, such as angiosarcoma, leiomyosarcoma, and ES, surgical resection remains the primary treatment when feasible. In cases where tumors are unresectable or metastatic, systemic chemotherapy and radiation therapy are considered. Liver transplantation is rarely indicated for hepatic sarcomas due to the high risk of recurrence. Recent advancements have introduced novel targeted therapies and immunotherapies for soft tissue sarcomas. For instance, tyrosine kinase inhibitor and angiogenesis inhibitor therapy are being invested in to treat angiosarcoma [273].

## 12. Future Perspectives

The diagnosis of hepatic mesenchymal tumors (HMTs) remains challenging due to overlapping morphological and immunohistochemical features. While current diagnostic approaches rely on morphology, IHC, and molecular analysis, a more comprehensive classification integrating pathological, immunological, and molecular characteristics is needed to enhance our understanding of HMT biology. The identification of highly specific biomarkers is crucial for improving early detection, prognostication, and therapeutic decision-making. Advances in molecular genetics continue to provide key insights into the pathogenesis, diagnosis, and potential targeted therapies for HMTs. The integration of these molecular and genetic findings into clinical practice can facilitate more precise, individualized treatment strategies, ultimately improving patient outcomes.

## 13. Conclusions

Mesenchymal tumors of the liver are much less common than epithelial tumors, but they represent a diverse group of neoplasms with distinct morphological and immunohistochemical characteristics. Most HMTs are not unique to the liver, with some exceptions, including MH, CNSEN, and ES. HMTs pose a diagnostic challenge due to their significant morphological overlap with other lesions and potential to mimic epithelial tumors. Recent advances in molecular pathology have identified specific genetic alterations in many of these mesenchymal tumors, allowing for accurate classification. Accurate diagnosis necessitates a comprehensive approach that includes detailed clinical and radiological assessments with thorough histological, immunohistochemical, and molecular analyses. As our understanding of the molecular basis of these tumors continues to advance through further study, diagnostic accuracy will improve, treatment strategies will be better guided, and patient outcomes will be enhanced.

## Figures and Tables

**Figure 1 biomedicines-13-00479-f001:**
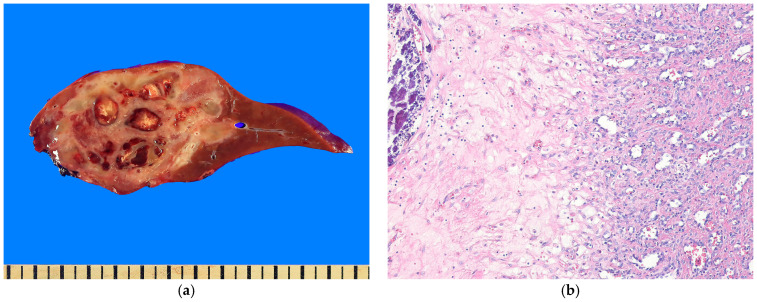
Hepatic congenital hemangioma. (**a**) The cut surface shows a white-tan mass with central calcification. (**b**) Capillary-type vascular channels lined by plump endothelial cells and less-cellular areas with fibrosis and calcification are present (hematoxylin-eosin stain, ×100).

**Figure 2 biomedicines-13-00479-f002:**
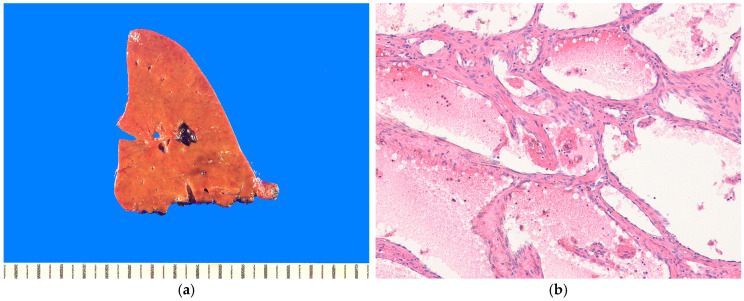
Cavernous hemangioma. (**a**) The cut surface shows a brown-tan cavitary lesion. (**b**) Dilated vascular spaces filled with blood are lined by flat endothelial cells and separated by fibrous septa (hematoxylin-eosin stain, ×100).

**Figure 3 biomedicines-13-00479-f003:**
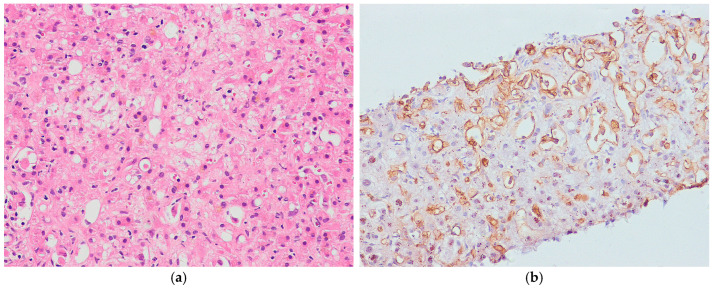
Epithelioid hemangioendothelioma. (**a**) The tumor cells contain intracytoplasmic vacuoles and infiltrate the sinusoidal spaces (hematoxylin-eosin stain, ×100). (**b**) The tumor cells are highlighted by CD31 (immunohistochemical stain for CD31, ×200).

**Figure 4 biomedicines-13-00479-f004:**
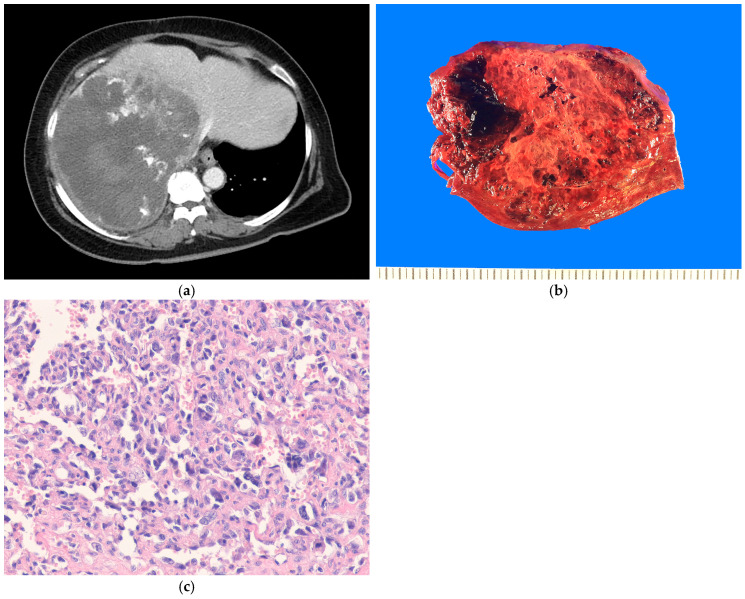
Angiosarcoma. (**a**) Abdominal computed tomography reveals a huge mass with hemorrhage in the right lobe. (**b**) The tumor shows a poorly defined, dark red brown mass with hemorrhage and necrosis. (**c**) Irregular vascular channels lined by atypical endothelial cells with hyperchromatic nuclei are present. Multilayering is present (hematoxylin-eosin stain, ×100).

**Figure 5 biomedicines-13-00479-f005:**
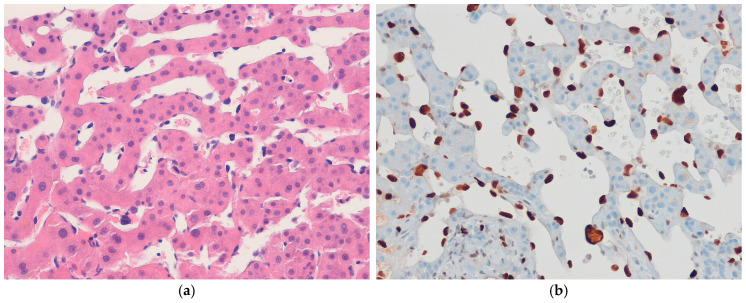
Sinusoidal-type angiosarcoma. (**a**) The dilated sinusoids are lined by atypical spindle and pleomorphic cells with hyperchromatic nuclei (hematoxylin-eosin stain, ×100). (**b**) These atypical cells are strongly positive for ERG (immunohistochemical stain for ERG, ×200).

**Figure 6 biomedicines-13-00479-f006:**
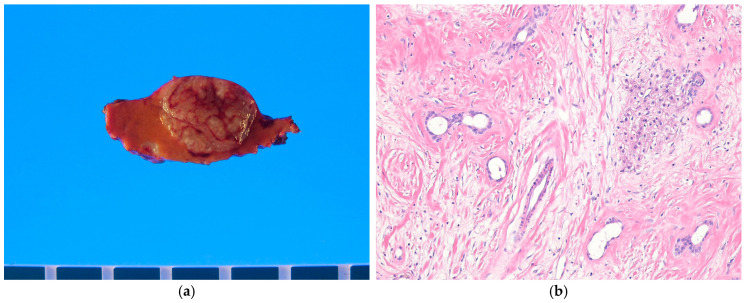
Mesenchymal hamartoma. (**a**) The cut surface shows a well-circumscribed, gray-tan solid mass. (**b**) Abundant mesenchymal stroma containing bland spindle-shaped cells, bile ducts with abnormal configuration surrounded by collagenous tissue, and islands of hepatocytes are present (hematoxylin-eosin stain, ×40).

**Figure 7 biomedicines-13-00479-f007:**
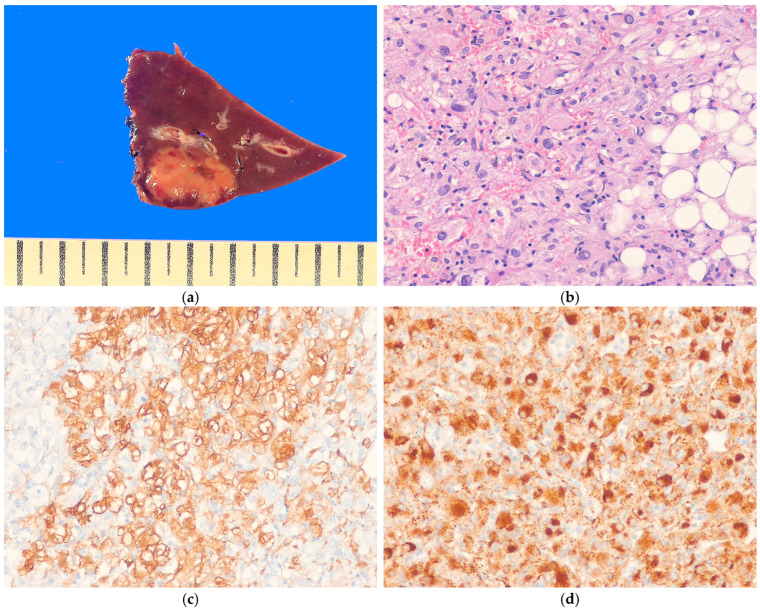
Perivascular epithelioid cell tumor (PEComa). (**a**) The cut surface shows a well-circumscribed, gray-tan solid tumor. (**b**) The tumor is composed of epithelioid cells with eosinophilic granular cytoplasm and adipocytes (hematoxylin-eosin stain, ×200). (**c**) The tumor cells are positive for smooth muscle actin (SMA) (immunohistochemical stain for SMA, ×200). (**d**) The tumor cells are also positive for HMB-45 (immunohistochemical stain for HMB-45, ×200).

**Figure 8 biomedicines-13-00479-f008:**
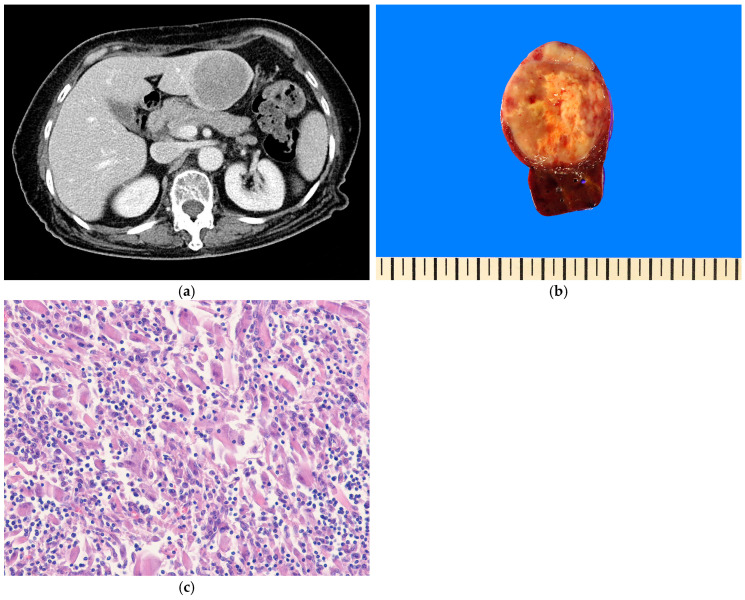
Inflammatory perivascular epithelioid cell tumor (PEComa). (**a**) Abdominal computed tomography reveals an early-enhancing hypodense mass in segment 3. (**b**) The tumor is well circumscribed, brown, yellow to gray, and solid. (**c**) The spindle and epithelioid tumor cells and a prominent inflammatory cell infiltrate, including lymphocytes and plasma cells, are present (hematoxylin-eosin stain, ×200).

**Figure 9 biomedicines-13-00479-f009:**
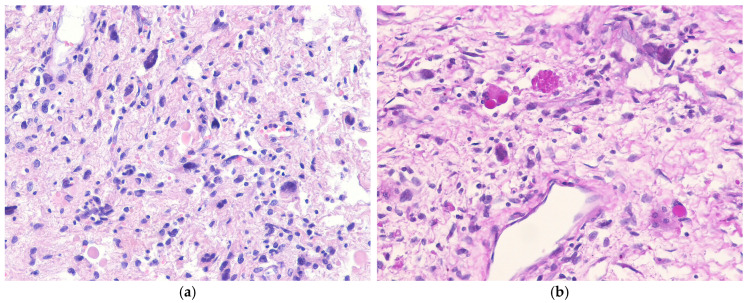
Embryonal sarcoma. (**a**) The tumor is composed of spindle and pleomorphic cells. Multinucleated giant cells and intracytoplasmic eosinophilic hyaline globules are present (hematoxylin-eosin stain, ×200). (**b**) These globules are periodic acid–Schiff (PAS)-positive and diastase-resistant (diastase periodic acid-Schiff stain, ×200).

**Table 1 biomedicines-13-00479-t001:** Relatively common and rare mesenchymal tumors of the liver.

**Adipocytic tumors**	**Pericytic (Perivascular) tumors**
*Benign*	*Benign*
Lipoma	Glomus tumor ^(b)^
*Malignant*	
Liposarcoma	**Smooth muscle tumors**
	Benign
**Fibroblastic and myofibroblastic tumors**	Leiomyoma
*Intermediate (rarely metastasizing)*	*Intermediate*
Inflammatory myofibroblastic tumor	EBV-associated smooth muscle tumor
Solitary fibrous tumor	*Malignant*
	Leiomyosarcoma
**Vascular tumors**	
*Benign*	**Skeletal muscle tumors**
Hepatic congenital hemangioma	*Malignant*
Hepatic infantile hemangioma	Embryonal rhabdomyosarcoma
Cavernous hemangioma	
Anastomosing hemangioma	**Tumors of uncertain differentiation**
Hepatic small vessel neoplasm ^(a)^	*Benign*
Lymphangioma	Mesenchymal hamartoma
*Intermediate (rarely metastasizing)*	PEComa/angiomyolipoma ^(b)^
Kaposi sarcoma	*Intermediate*
*Malignant*	Calcifying nested stromal–epithelial tumor
Epithelioid hemangioendothelioma	*Malignant*
Angiosarcoma	Embryonal sarcoma
	Extrarenal rhabdoid tumor

^(a)^ Hepatic small vessel neoplasm is a benign or low-grade neoplasm. ^(b)^ Glomus tumor and PEComa are most commonly benign, although they can rarely be malignant. EBV; Epstein–Varr virus; PEComa, perivascular epithelioid cell tumor.

**Table 2 biomedicines-13-00479-t002:** Clinical and histological features, immunohistochemical markers, and molecular features of selected hepatic mesenchymal tumors.

Tumor Category	Tumor Type	Clinical Features	Histological Features	IHC Markers	Molecular Features
Fibroblastic and myofibroblastic tumors	Inflammatory myofibroblastic tumor	Children and young adults are most often affected; abdominal pain, abdominal discomfort, fever, and weight loss	Loose or compact fascicles of plump spindle cells with inflammatory infiltrate of lymphocytes and plasma cells	Variably positive for SMA, desmin, and CKs; positive for ALK (50%), ROS1 (5%)	*ALK* rearrangements with various partner genes; *ROS1*, *PDGFRB*, and *RET* rearrangements
	Solitary fibrous tumor	Usually adults; more frequent in women; abdominal mass, abdominal pain, hypoglycemia	Ovoid-to-spindle cells arranged in short or haphazard fascicles; prominent branching, staghorn vasculature	Positive for CD34 and STAT6	*NAB2*::*STAT6*
Vascular tumors	Hepatic congenital hemangioma	Newborns, and infants; hepatomegaly, thrombocytopenia, hypofibrinogenemia	Aggregates and lobules of capillary-type channels; fibrous stroma; dilated hepatic sinusoids	Positive for vascular markers ^(a)^; negative for GLUT1	*GNAQ* and *GNA11* mutations; *PIK3CA* mutations (a subset)
	Hepatic infantile hemangioma	Infants, children; female predominance; premature, of low birth weight; cutaneous hemangioma	Anastomosing, sinusoid-like channels lined by mostly flattened and bland endothelial cells	Positive for vascular markers; positive for GLUT1	Activation of hedgehog signaling pathway components
	Cavernous hemangioma	Any age; more common in women; larger tumors may be symptomatic; grow in estrogen exposure	Blood-filled, dilated vascular channels lined by bland endothelial cells with fibrous septa	Positive for vascular markers	No distinct findings to date
	Anastomosing hemangioma	Adults; more common in women; well-circumscribed lesions;	Anastomosing capillary-like vascular channels lined by a single layer of endothelial cells; lobular architecture	Positive for vascular markers	*GNAQ*, *GNA11*, and *GNA14* mutations
	Hepatic small vessel neoplasm	Adults; more common in men; benign or low-grade; asymptomatic single liver mass	Small and poorly defined; infiltrative proliferation of small, thin-walled vessels lined by flattened-to-plump, hobnail endothelial cells	Positive for vascular markers; Ki-67 index <10%	*GNAQ*, *GNA11*, and *GNA14* mutations
	Kaposi sarcoma	Any age; HHV8 infection; occur in immunosuppressed patients; HIV-associated	Small, irregular vascular channels and fascicles of uniform spindle endothelial cells with mild atypia	Positive for vascular markers; positive for HHV8	Recurrent 11q13 gains with *FGF4* and *INT2* up-regulation
	Epithelioid hemangio- endothelioma	Middle-aged adults; slight female predominance; abdominal pain, weight loss, and ascites	Cords of the epithelioid and cells within a myxohyaline or fibrous stroma; cytoplasmic vacuolization	Positive for vascular markers; positive for CAMTA1 or TFE3	*WWTR1*::*CAMTA1* (>90%) *YAP1*::*TFE3* (5%)
	Angiosarcoma	Early childhood, adults; risk factors such as chemical carcinogens (vinyl chloride, androgens), radiation	Anastomosing vascular channels with moderate to marked endothelial atypia; multilayering; conspicuous mitotic activity; necrosis	Positive for vascular markers. Ki-67 index > 10%; frequently positive for p53	Complex genomic profiles; loss of ATRX, with an associated alternative lengthening of telomeres
Pericytic tumors	Glomus tumor	Adults; male predominance; larger than cutaneous lesions; epigastric fullness, pain	Uniform, round cells with well-defined cell borders; arranged in perivascular nests	Positive for SMA, and h-caldesmon; pericellular positive for collagen type IV	NOTCH family gene rearrangements; *BRAF* p.V600E mutations
Smooth muscle tumors	EBV-associated smooth muscle tumor	Wide age range; three groups: HIV-associated, post-transplant, congenital/primary immunodeficiency	Fascicles of spindle cells with blunt-ended nuclei and eosinophilic cytoplasm; more primitive round cells	Positive for SMA, h-caldesmon, and desmin; positive for EBER ISH	MYC overexpression; AKT/mTOR pathway activation
	Leio- myosarcoma	Adults; immunosuppressed children; metastatic leiomyosarcomas are more common than primary	Fascicles of spindle cells; epithelioid, pleomorphic cells; nuclear atypia, increased mitoses, necrosis	Positive for SMA, h-caldesmon, and desmin	Complex aneuploid karyotypes; inactivation of *TP53* and *RB1*
Skeletal muscle tumors	Embryonal rhabdo-myosarcoma	Most patients are under 5 years of age; commonly arise in the biliary tracts; obstructive jaundice	Primitive round and spindle cells with scattered rhabdomyoblasts; cambium layer	Positive for desmin, myogenin, and MYOD1	Chromosomal gain (2,8,11,12,13); LOH at 11p15.5
Tumors of uncertain differentiation	Mesenchymal hamartoma	Most cases before the age of 3 years; <5% after the age of 5 years; slightly elevated serum AFP	Abundant mesenchymal stroma; bile ducts with abnormal configuration; islands of hepatocytes	No specific markers	Chromosomal rearrangements involving 19q13.4
	PEComa/ angiomyolipoma	Middle-aged adults; female predominance; associated with tuberous sclerosis (5–10%)	Nests of epithelioid or spindle cells with granular eosinophilic-to-clear cytoplasm; adipocytes and thick-walled blood vessels	Positive for smooth muscle markers and melanocytic markers ^(b)^; positive for TFE3 (subset)	*TSC1* or *TSC2* mutations; *TFE3* rearrangements (subset)
	Calcifying nested stromal–epithelial tumor	Children, adolescents; young adults; Cushing syndrome (20%); often history of a calcified hepatic nodule	Nested architecture of bland and uniform epithelioid and spindle cells; surrounded by cellular myofibroblastic stroma	Positive for CKs, WT1; nuclear β-catenin staining	*CTNNB1* deletions
	Embryonal sarcoma	Children aged 5–15 years; abdominal distension, pain, fever; leukocytosis, elevated serum ALP levels	Spindled, stellate, and pleomorphic giant cells in a myxoid stroma; Giant cells; intracytoplasmic globules	Lack of specific markers; variably positive for CKs, desmin, and SMA	Chromosomal rearrangements involving 19q13.4; *TP53* mutations
	Extrarenal rhabdoid tumor	Before the age of 3 years; abdominal mass, distention, pain; tumor rupture	Rhabdoid cells with large vesicular rounded nuclei, prominent nucleoli, and abundant cytoplasm	Loss of SMARCB1 (INI1) expression	Mutations or deletions in *SMARCB1 (INI1)*

^(a)^ Vascular markers include CD31, CD34, factor VIII-related antigen, ERG, and FLI1; ^(b)^ melanocytic markers include HMB-45, melan-A, and MITF; IHC, immunohistochemistry; EBV, Epstein–Barr virus; ISH, in situ hybridization; HHV8, human herpesvirus 8; HIV, human immunodeficiency virus; SMA, smooth muscle actin; EBER ISH, Epstein–Barr virus-encoded small RNA in situ hybridization; INI1, integrase interactor 1; AFP; alpha-fetoprotein; PEComa, perivascular epithelioid tumor; CK, cytokeratin; TFE3, transcription factor E3; ALP, alkaline phosphatase.

**Table 3 biomedicines-13-00479-t003:** Cell morphology and additional findings of selected hepatic mesenchymal tumors.

Cell Morphology	Tumor Types
Spindle cell	Solitary fibrous tumor Inflammatory myofibroblastic tumor Kaposi sarcoma Angiosarcoma Leiomyosarcoma
Round cell	Myxoid liposarcoma Embryonal rhabdomyosarcoma
Epithelioid cell	Epithelioid hemangioendothelioma Epithelioid angiosarcoma PEComa/angiomyolipoma
Pleomorphic cell	Dedifferentiated liposarcoma Pleomorphic leiomyosarcoma Embryonal sarcoma
Adipocytic component	Lipoma Liposarcoma Fat-forming solitary fibrous tumor PEComa/angiomyolipoma
Vasoformative pattern	Hemangioma Angiosarcoma
Biphasic pattern	Calcifying nested stromal–epithelial tumor
Prominent inflammatory cells	Inflammatory myofibroblastic tumor Inflammatory PEComa

PEComa: perivascular epithelioid cell tumor.

**Table 4 biomedicines-13-00479-t004:** Immunohistochemistry of selected hepatic mesenchymal tumors.

	LPS	SFT	IMT	EHE	AS	LMS	ERMS	PEC	ES	ERT
SMA	− ^(a)^	+/−	+/−	−	−	+	+/−	+	+/−	−
Desmin	− ^(a)^	−	+/−	−	−	+	+	+/−	+/−	−
Myogenin	−	−	−	−	−	−	+	−	+	−
CD31	−	−	−	+	+	−	−	−	−	−
CD34	−	+	+/−	+	+	−	−	−	−	−
ERG	−	−	−	+	+	−	−	−	−	−
CAMTA-1	−	−	−	+	−	−	−	−	−	−
MDM2	+ ^(b)^	−	−	−	−	−	−	−	−	−
STAT6	−	+ ^(c)^	−	−	−	−	−	−	−	−
ALK	−	−	+ ^(d)^	−	−	−	−	−	−	−
INI1	Retained	Retained	Retained	Retained	Retained	Retained	Retained	Retained	Retained	Loss
HMB-45	−	−	−	−	−	−	−	+	−	−
Cytokeratins	−	−	+/−	+/−	+/−	+/−	−	−	−	+

^(a)^ May be expressed in a subset of dedifferentiated liposarcoma. ^(b)^ Expressed in well-differentiated and dedifferentiated liposarcoma. ^(c)^ Decreased or lost in dedifferentiated solitary fibrous tumor. ^(d)^ Expressed in approximately 50% of inflammatory myofibroblastic tumor. +, positive staining; +/−, focal or variable staining; −, negative staining; LPS, liposarcoma; SFT, solitary fibrous tumor; IMT, inflammatory myofibroblastic tumor; EHE, epithelioid hemangioendothelioma; AS, angiosarcoma; LMS, leiomyosaromca; ERMS, embryonal rhabdomyosarcoma; PEC, perivascular epithelioid cell tumor; ES, embryonal sarcoma; ERT, extrarenal rhabdoid tumor; SMA, smooth muscle actin; ERG, ETS-related gene; CAMTA-1, calmodulin-binding transcription activator 1; MDM2, mouse double minute 2 homolog; STAT6, signal transducer and activator of transcription 6; ALK, anaplastic lymphoma kinase; INI1, integrase interactor 1; HMB-45, human melanoma black-45.

## Data Availability

All data were included in the manuscript.

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
