# Peer review of "Mesenchymal Tumors of the Liver: An Update Review"

_biomedicines, 2025, doi:10.3390/biomedicines13020479_

Round 1

Reviewer 1 Report

Comments and Suggestions for Authors

This is a well-written, informative review of hepatic mesenchymal tumors. The authors focused their interest on these tumors' diagnosis and pathological features.

Suggested changes:

1)        Although it is not mandatory, the manuscript could be improved by adding some epidemiological information before describing each tumor.

2)        Moreover, adding a final section before the Conclusion with a brief description of the currently used treatment of these tumors would also enhance the quality of this review.

3)        For increased gene expression, please consider using up-regulation instead of amplification.

4)        Be consistent in using italic font for gene names throughout the manuscript.

5)        Please correct the heading 4.9. Angiosacroma.

Author Response

This is a well-written, informative review of hepatic mesenchymal tumors. The authors focused their interest on these tumors' diagnosis and pathological features.

Suggested changes:

1) Although it is not mandatory, the manuscript could be improved by adding some epidemiological information before describing each tumor.

-> We appreciate your valuable suggestion. We have incorporated relevant epidemiological information, including incidence, age distribution, and gender predominance, where applicable.

-> In the section on Kaposi Sarcoma,

. . . Four clinical and epidemiological forms are known: (1) classic indolent KS, (2) endemic African KS, (3) acquired immunodeficiency syndrome (AIDS)-associated KS, and (4) iatrogenic KS. . . Some hepatic KS cases may be associated with iatrogenic factors, such as immunosuppression following transplantation.

-> In the section on Hepatic Infantile Hemangioma,

. . Many patients are asymptomatic. When present, symptoms are nonspecific and may include nausea, lethargy, or gastrointestinal bleeding.

-> In the section on Hepatic Infantile Hemangioma,

Approximately 70% of patients have elevated alkaline phosphatase levels [119].

-> In the section on Mesenchymal Hamartoma

. . Symptoms may include an enlarging abdomen and a nontender mass.

-> In the section on PEComas,

Large lesions may present with abdominal pain.

2) Moreover, adding a final section before the Conclusion with a brief description of the currently used treatment of these tumors would also enhance the quality of this review.

-> Thank you for your comments. We have added a new section before the Conclusion to discuss the treatment strategies for hepatic mesenchymal tumors (HMTs) as follows.

  1. Treatment of Hepatic Mesenchymal Tumors

Treatment strategies of HMTs vary depending on tumor type, size, growth pattern,  and malignancy potential. Small, asymptomatic hemangiomas (≤5 cm) require no treatment. However, surgical resection is recommended for larger hemangiomas (> 10 cm) and when patients present with symptoms such as pain, bleeding, or compression of adjacent structures. For hepatic sarcomas, such as angiosarcoma, leiomyosarcoma, and ES, surgical resection remains the primary treatment when feasible. In cases where tumors are unresectable or metastatic, systemic chemotherapy and radiation therapy are considered. Liver transplantation is rarely indicated for hepatic sarcomas due to the high risk of recurrence. Recent advancements have introduced novel targeted therapies and immunotherapies for soft tissue sarcomas. For instance, tyrosine kinase inhibitors and angiogenesis inhibitors therapy are being invested in to treat angiosarcoma [277].

3) For increased gene expression, please consider using up-regulation instead of amplification.

-> Thank you for your suggestion. We have revised the manuscript accordingly and replaced "amplification" with "up-regulation" to describe increased gene expression more accurately.

4) Be consistent in using italic font for gene names throughout the manuscript.

-> We appreciate your suggestions. We have ensured that all gene names are consistently italicized throughout the manuscript, including both the main text and references.

 -> In references,

Reference No. 99. Bean, G.R…..Recurrent GNAQ mutations in

Reference No. 218. Huang, S.C… of TSC1/TSC2

Reference No. 222. McCarthy, .. of TP53, RB1, and ATRX.

5) Please correct the heading 4.9. Angiosacroma.

-> Thank you for your comments. We have corrected the heading as follows:

4.9. Angiosarcoma

Reviewer 2 Report

Comments and Suggestions for Authors

This paper provides a comprehensive and well-structured review of HMTs, detailing their clinical features, pathological characteristics, and diagnostic challenges. The manuscript is well-written and informative, with a strong foundation in recent literature and diagnostic advancements. However, several issues need to be addressed to improve clarity and strengthen the comprehensiveness of this study. 

Comments:

1.    The review would benefit from a dedicated section on clinical management and treatment strategies for HMTs, if applicable. While diagnosis is well covered, therapeutic approaches are underexplored.

2.    Consider including the emerging molecular and genetic insights that might impact future diagnostic or therapeutic strategies.

3.    Some sentences are overly complex and could be simplified for better readability. Example: “Accurate pathological diagnosis and a thorough understanding of the molecular pathogenesis of HMTs are critical for effective patient management and prognosis prediction.” can be simplified as “Accurate pathological diagnosis and understanding of HMTs’ molecular pathogenesis are crucial for patient management and prognosis.”

4.    While the manuscript presents detailed pathological features, there is limited discussion on the clinical implications of these tumors.

Author Response

This paper provides a comprehensive and well-structured review of HMTs, detailing their clinical features, pathological characteristics, and diagnostic challenges. The manuscript is well-written and informative, with a strong foundation in recent literature and diagnostic advancements. However, several issues need to be addressed to improve clarity and strengthen the comprehensiveness of this study.

Comments:

  1. The review would benefit from a dedicated section on clinical management and treatment strategies for HMTs, if applicable. While diagnosis is well covered, therapeutic approaches are underexplored.

-> Thank you for your comments. We have added a new section before the Conclusion to address treatment strategies for hepatic mesenchymal tumors (HMTs) as follows:

  1. Treatment of Hepatic Mesenchymal Tumors

Treatment strategies of HMTs vary depending on tumor type, size, growth pattern,  and malignancy potential. Small, asymptomatic hemangiomas (≤5 cm) require no treatment. However, surgical resection is recommended for larger hemangiomas (> 10 cm) and when patients present with symptoms such as pain, bleeding, or compression of adjacent structures. For hepatic sarcomas, such as angiosarcoma, leiomyosarcoma, and ES, surgical resection remains the primary treatment when feasible. In cases where tumors are unresectable or metastatic, systemic chemotherapy and radiation therapy are considered. Liver transplantation is rarely indicated for hepatic sarcomas due to the high risk of recurrence. Recent advancements have introduced novel targeted therapies and immunotherapies for soft tissue sarcomas. For instance, tyrosine kinase inhibitors and angiogenesis inhibitors therapy are being invested in to treat angiosarcoma [277].

  1. Consider including the emerging molecular and genetic insights that might impact future diagnostic or therapeutic strategies.

-> Thank you for your comments. We have added a Future Perspectives section before the Conclusion, incorporating emerging molecular and genetic insights, as follows.

  1. Future Perspectives

The diagnosis of hepatic mesenchymal tumors (HMTs) remains challenging due to overlapping morphological and immunohistochemical features. While current diagnostic approaches rely on morphology, IHC, and molecular analysis, a more comprehensive classification integrating pathological, immunological, and molecular characteristics is needed to enhance our understanding of HMTs biology. The identification of highly specific biomarkers is crucial for improving early detection, prognostication, and therapeutic decision-making. Advances in molecular genetics continue to provide key insights into the pathogenesis, diagnosis, and potential targeted therapies for HMTs. The integration of these molecular and genetic findings into clinical practice can facilitate more precise, individualized treatment strategies, ultimately improving patient outcomes.

  1. Some sentences are overly complex and could be simplified for better readability. Example: “Accurate pathological diagnosis and a thorough understanding of the molecular pathogenesis of HMTs are critical for effective patient management and prognosis prediction.” can be simplified as “Accurate pathological diagnosis and understanding of HMTs’ molecular pathogenesis are crucial for patient management and prognosis.”

-> Thank you for your valuable comments. We have revised the sentence for improved readability as follows:

-> “Accurate pathological diagnosis and a thorough understanding of the molecular pathogenesis of HMTs are critical for effective patient management and prognosis prediction.” -> Accurate pathological diagnosis and understanding of HMTs’ molecular pathogenesis are crucial for patient management and prognosis.

-> “their own chapter” -> their chapter

-> a strong association with hepatic steatosis -> strongly associated with hepatic steatosis

-> an aggressive subtype of IMT -> an aggressive IMT subtype

-> The majority of pediatric cases -> most pediatric cases

-> a solid sheet pattern -> solid sheets

-> to make a definitive diagnosis -> to diagnose definitively

  1. While the manuscript presents detailed pathological features, there is limited discussion on the clinical implications of these tumors.

-> We appreciate your valuable suggestion. We have discussed clinical implications in Kaposi sarcoma, hepatic infantile hemangioma, epithelioid hemangioendothelioma, and mesenchymal hamartoma.

-> In the section on Kaposi Sarcoma,

. . . Four clinical and epidemiological forms are known: (1) classic indolent KS, (2) endemic African KS, (3) acquired immunodeficiency syndrome (AIDS)-associated KS, and (4) iatrogenic KS. . . Some hepatic KS cases may be associated with iatrogenic factors, such as immunosuppression following transplantation.

-> In the section of Hepatic Infantile Hemangioma,

. . Many patients are asymptomatic. When present, symptoms are nonspecific and may include nausea, lethargy, or gastrointestinal bleeding.

-> In the section of Hepatic Infantile Hemangioma,

Approximately 70% of patients have elevated alkaline phosphatase levels [119].

-> In the section of Mesenchymal Hamartoma

. . Symptoms may include an enlarging abdomen and a nontender mass

-> In the section of PEComa,

. . Large lesions may present with abdominal pain.

Reviewer 3 Report

Comments and Suggestions for Authors

Mesenchymal tumors of the liver are very rare. This article is very interesting, but there are some problems and is not acceptable for publication in the present form.

Major points

1)   Clinical features: Too stereotype. Please discuss more deeply.

2)   Differential diagnosis: Always too short. Please add medical imaging findings (CT, MR, and contrast-enhanced sonography). If not, the authors’ description remains too superficial.

3)   Figures: Too few. Please add pathologic pictures of other tumors. If possible, please add their medical imaging findings.

Minor points

1)   Please add a list of abbreviations.

2)   L.21. In my opinion, portovenous shunting does not cause cardiac failure.

3)   PE coma: Please discuss two groups (angiomyolipoma and non-angiomyolipoma) separately.

Author Response

Mesenchymal tumors of the liver are very rare. This article is very interesting, but there are some problems and is not acceptable for publication in the present form.

Major points

1) Clinical features: Too stereotype. Please discuss more deeply.

-> Thank you for your comments. We agree with your suggestions and have expanded the discussion on the clinical features and imaging characteristics of HMTs.

2) Differential diagnosis: Always too short. Please add medical imaging findings (CT, MR, and contrast-enhanced sonography). If not, the authors’ description remains too superficial.

-> Thank you for your valuable comments. We have expanded the paragraph on differential diagnosis to include characteristic radiological imaging findings of selected HMTs in the Diagnostic Approach section.

  The characteristic radiological findings are helpful for the differential diagnosis of selected HMTs. HIH typically appears as a hypoechoic or complex mass on ultrasonog-raphy [276]. Non-enhanced CT scans reveal a low-attenuation mass, with fine calcifica-tions present in approximately 50% of cases. MRI is particularly useful for assessing multifocality. Arteriography may reveal enlarged, tortuous feeding arteries. Cavernous hemangiomas are isodense to large blood vessels and exhibit characteristic cloud-like peripheral enhancement on CT. EHE is characterized by low-attenuation peripheral tumor nodules on CT. MRI often demonstrates a signal halo around the nodules. Angiosarcomas present as hypodense masses on nonenhanced CT, which become isodense on delayed postcontrast scans. Angiography may reveal an abnormal vascular pattern. MH appears to be a complex multicystic mass on ultrasonography. CT imaging reveals variability in cyst size and septal thickness. AML appears as a heterogeneous, well-circumscribed, hyperechoic lesion on ultrasound. On CT, it presents as a hypo-dense mass with marked early contrast enhancement and delayed enhancement in the portal venous phase. MRI is highly sensitive for detecting the fatty component, which exhibits high signal intensity on T1-weighted images. ES is characterized by a hypo-dense mass with solid and cystic areas on CT. It is typically hypovascular, distinguishing it from other vascular tumors.

3) Figures: Too few. Please add pathologic pictures of other tumors. If possible, please add their medical imaging findings.

-> Thank you for your comments. We agree with your suggestion.

We have added microscopic and imaging photographs for selected HMTs, including epithelioid hemangioendothelioma, inflammatory PEComa, and angiosarcoma.

Minor points

1) Please add a list of abbreviations.

-> Thank you for your comments. We have added a list of abbreviations.

2) L.21. In my opinion, portovenous shunting does not cause cardiac failure.

-> Thank you for your comments. We have removed “portohepatic venous.”

as a result of arteriovenous or portohepatic venous shunting.

   -> as a result of arteriovenous shunting.

3) PEComa: Please discuss two groups (angiomyolipoma and non-angiomyolipoma) separately.

-> Thank you for your comments. We have revised the manuscript to explicitly describe the two main groups of PEComas: angiomyolipomas (AMLs) and non-angiomyolipoma PEComas.

-> PEComas are generally classified into two main groups: (1) angiomyolipomas (AMLs) and (2) non-angiomyolipoma PEComas. AML is a subtype of PEComa that typically arises in the kidney and liver. It is composed of varying proportions of blood vessels, smooth muscle, and adipose tissue, leading to triphasic morphology. Non-angiomyolipoma PEComas can arise in various locations, including soft tissue, the uterus, lungs, and gastrointestinal tract. Unlike AMLs, they lack triphasic histologic features and are often composed predominantly of spindle or epithelioid cells with perivascular accentuation. These tumors share the same perivascular epithelioid cell lineage as AML.

Round 2

Reviewer 2 Report

Comments and Suggestions for Authors

All my concerns have been addressed.